# HARNESSING OUT-OF-DISTRIBUTION EXAMPLES VIA AUGMENTING CONTENT AND STYLE

**Zhuo Huang[1], Xiaobo Xia[1], Li Shen[2], Bo Han[3],**
**Mingming Gong[4], Chen Gong[5,\*], Tongliang Liu[1,\*]**

[1]Syndey AI Centre, The University of Sydney; [2]JD Explore Academy;
[3]Department of Computer Science, Hong Kong Baptist University; [4]The University of Melbourne;
[5]Key Laboratory of Intelligent Perception and Systems for High-Dimensional Information
of Ministry of Education, School of Computer Science and Engineering, Nanjing University of Science and Technology
`{zhua0420, xxia5420}@uni.sydney.edu.au, mathshenli@gmail.com,`
`bhanml@comp.hkbu.edu.hk, mingming.gong@unimelb.edu.au,`
`chen.gong@njust.edu.cn, tongliang.liu@sydney.edu.au`

## ABSTRACT

Machine learning models are vulnerable to Out-Of-Distribution (OOD) examples, and such a problem has drawn much attention. However, current methods lack a full understanding of different types of OOD data: there are *benign* OOD data that can be properly adapted to enhance the learning performance, while other *malign* OOD data would severely degenerate the classification result. To Harness OOD data, this paper proposes a HOOD method that can leverage the *content* and *style* from each image instance to identify benign and malign OOD data. Particularly, we design a variational inference framework to causally disentangle content and style features by constructing a structural causal model. Subsequently, we augment the content and style through an intervention process to produce malign and benign OOD data, respectively. The benign OOD data contain novel styles but hold our interested contents, and they can be leveraged to help train a style-invariant model. In contrast, the malign OOD data inherit unknown contents but carry familiar styles, by detecting them can improve model robustness against deceiving anomalies. Thanks to the proposed novel disentanglement and data augmentation techniques, HOOD can effectively deal with OOD examples in unknown and open environments, whose effectiveness is empirically validated in three typical OOD applications including OOD detection, open-set semi-supervised learning, and open-set domain adaptation.

## 1 INTRODUCTION

Learning in the presence of Out-Of-Distribution (OOD) data has been a challenging task in machine learning, as the deployed classifier tends to fail if the unseen data drawn from unknown distributions are not properly handled (Hendrycks & Gimpel, 2017; Pan & Yang, 2009). Such a critical problem ubiquitously exists when deep models meet domain shift (Ganin et al., 2016; Tzeng et al., 2017) and unseen-class data (Hendrycks & Gimpel, 2017; Scheirer et al., 2012), which has drawn a lot of attention in some important fields such as OOD detection (Hein et al., 2019; Hendrycks & Gimpel, 2017; Lee et al., 2018; Liang et al., 2018; Liu et al., 2020; Wang et al., 2022a; 2023; 2022b), Open-Set Domain Adaptation (DA) (Liu et al., 2019; Saito et al., 2018), and Open-Set Semi-Supervised Learning (SSL) (Huang et al., 2021b; 2022b;a; Oliver et al., 2018; Saito et al., 2021; Yu et al., 2020).

In the above fields, OOD data can be divided into two types, namely *benign* OOD data[1] and *malign* OOD data. The benign OOD data can boost the learning performance on the target distribution through DA techniques (Ganin & Lempitsky, 2015; Tzeng et al., 2017), but they can be misleading if not being properly exploited. To improve model generalization, many *positive data augmentation* techniques (Cubuk et al., 2018; Xie et al., 2020) have been proposed. For instance, the performance of SSL (Berthelot et al., 2019; Sohn et al., 2020) has been greatly improved thanks to the augmented benign OOD data. On the contrary, malign OOD data with unknown classes can damage the

---

*Corresponding to Tongliang Liu and Chen Gong.

[1]We follow (Bengio et al., 2011) to regard the augmented data as a type of OOD data

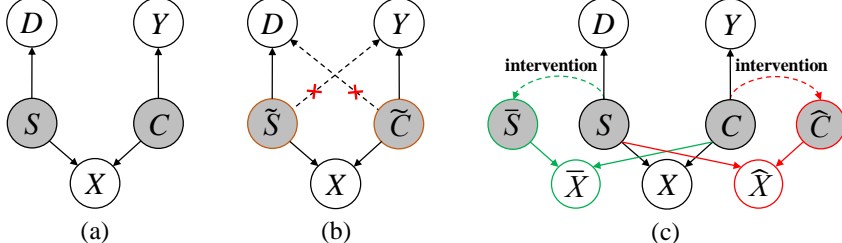

Figure 1: (a) An **ideal** causal diagram which reveals the data generating process. (b) Illustration of our disentanglement. The brown-edged variables $\tilde{C}$ and $\tilde{S}$ are approximations of content $C$ and style $S$. The dashed lines indicate the unwanted causal relations that should be broken. (c) Illustration of the data augmentation of HOOD. The green lines and red lines denote the augmentation of benign OOD data $\bar{X}$ and malign OOD data $\hat{X}$, respectively. In all figures, the blank variables are observable and the shaded variables are latent.

classification results, but they are deceiving and hard to detect (Hendrycks & Gimpel, 2017; Liang et al., 2018; Wei et al., 2022b;a). To train a robust model against malign OOD data, some works (Kong & Ramanan, 2021; Sinha et al., 2020) conduct *negative data augmentation* to generate "hard" malign data which resemble in-distribution (ID) data. By separating such "hard" data from ID data, the OOD detection performance can be improved. When presented with both malign and benign OOD data, it is more challenging to decide which to separate and which to exploit. As a consequence, the performance of existing open-set methods could be sub-optimal due to two drawbacks: 1) radically exploiting too much malign OOD data, and 2) conservatively denying too much benign OOD data.

In this paper, we propose a HOOD framework (see Fig. 2) to properly harness OOD data in several OOD problems. To distinguish benign and malign OOD data, we model the data generating process by following the structural causal model (SCM) (Glymour et al., 2016; Pearl, 2009; Gao et al., 2022) in Fig. 1 (a). Particularly, we decompose an image instance $X$ into two latent components: 1) *content* variable $C$ which denotes the interested object, and 2) *style* variable $S$ which contains other influential factors such as brightness, orientation, and color. The content $C$ can indicate its true *class* $Y$, and the style $S$ is decisive for the environmental condition, which is termed as *domain* $D$. Intuitively, malign OOD data cannot be incorporated into network training, because they contain unseen contents, thus their true classes are different from any known class; and benign OOD data can be adapted because they only have novel styles but contain the same contents as ID data. Therefore, we can distinguish the benign and malign OOD data based on the extracted the content and style features.

In addition, we conduct causal disentanglement through maximizing an approximated evidence lower-bound (ELBO) (Blei et al., 2017; Yao et al., 2021; Xia et al., 2022b) of joint distribution $P(X, Y, D)$. As a result, we can effectively break the spurious correlation (Pearl, 2009; Glymour et al., 2016; Hermann et al., 2020; Li et al., 2021b; Zhang et al., 2022) between content and style which commonly occurs during network training (Arjovsky et al., 2019), as shown by the dashed lines in Fig. 1 (b). In the ablation study, we find that HOOD can correctly disentangle content and style, which can correspondingly benefit generalization tasks (such as open-set DA and open-set SSL) and detection task (such as OOD detection).

To further improve the learning performance, we conduct both positive and negative data augmentation by solely intervening the style and content, respectively, as shown by the blue and red lines in Fig. 1 (c). Such process is achieved through backpropagating the gradient computed from an intervention objective. As a result, style-changed data $\bar{X}$ must be identified as benign OOD data, and content-changed data $\hat{X}$ should be recognized as malign OOD data. Without including any bias, the benign OOD data can be easily harnessed to improve model generalization, and the malign OOD data can be directly recognized as harmful ones which benefits the detection of unknown anomalies. By conducting extensive experiments on several OOD applications, including OOD detection, open-set SSL, and open-set DA, we validate the effectiveness of our method on typical benchmark datasets. To sum up, our contributions are three-fold:

- We propose a unified framework dubbed HOOD which can effectively disentangle the content and style features to break the spurious correlation. As a result, benign OOD data and malign OOD data can be correctly identified based on the disentangled features.

- We design a novel data augmentation method which correspondingly augments the content and style features to produce benign and malign OOD data, and further leverage them to enhance the learning performance.

- We experimentally validate the effectiveness of HOOD on various OOD applications, including OOD detection, open-set SSL, and open-set DA.

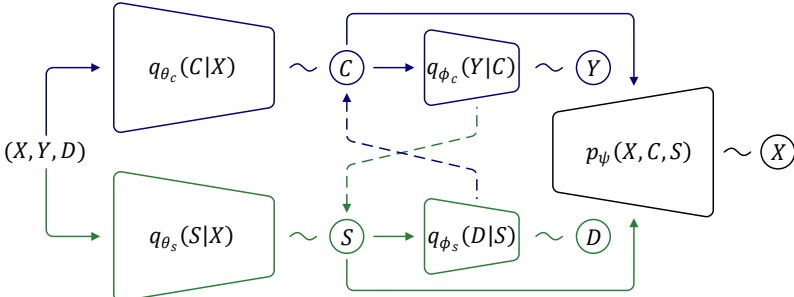

Figure 2: Architecture of the HOOD. The solid lines denote the inference flow, the dashed lines indicate the disentanglement of content and style, and the tildes stand for the approximation of the corresponding variables.

## 2 METHODOLOGY

In this section, we propose our HOOD framework as shown in Fig. 2. Specifically, we utilize the class labels of labeled data and the pseudo labels (Lee, 2013) of unlabeled data as the class supervision to capture the content feature. Moreover, we perform different types of data augmentation and regard the augmentation types as the domain supervision for each style. Thereby, each instance $\mathbf{x}$ is paired with a class label $y$ and a domain label $d$. Then, we apply two separate encoders $g_c$ and $g_s$ parameterized by $\theta_c$ and $\theta_s$ to model the posterior distributions $q_{\theta_c}(C \mid X)$ and $q_{\theta_s}(S \mid X)$, respectively. Subsequently, the generated $C$ and $S$ are correspondingly fed into two fully-connected classifiers $f_c$ and $f_s$ parameterized by $\phi_c$ and $\phi_s$, which would produce the label predictions $q_{\phi_c}(Y \mid C)$ and $q_{\phi_s}(D \mid S)$, respectively. To further enhance the identifiability of $C$ and $S$, a decoder $h$ with parameter $\psi$ is employed to reconstruct the input instance $\mathbf{x}$ based on its content and style.

Below, we describe the detailed procedures and components during modeling HOOD. We first introduce the proposed variational inference framework for disentangling the content and style based on the constructed SCM. Subsequently, we conduct intervention to produce benign OOD data and malign OOD data. Further, we appropriately leverage the benign and malign OOD data to boost the learning performance. Finally, we formulate the deployment of HOOD in three OOD applications.

### 2.1 VARIATIONAL INFERENCE FOR CONTENT AND STYLE DISENTANGLEMENT

First, we assume that the data generating process can be captured by certain probability distributions. Therefore, according to the constructed SCM in Fig. 1 (a), the joint distribution $P(X, Y, D, C, S)$ of the interested variables can be factorized as follows:

$$P(X, Y, D, C, S) = P(C, S)P(Y, D \mid C, S)P(X \mid C, S). \tag{1}$$

Based on the SCM in Fig. 1 (a), $Y$ and $D$ are conditionally independent to each other, i.e., $Y \perp\!\!\!\perp D \mid (C, S)$, so we have $P(Y, D \mid C, S) = P(Y \mid C, S)P(D \mid C, S)$. Similarly, we have $P(C, S) = P(C)P(S)$. Moreover, we can also know that $Y$ is not conditioned on $S$, and $D$ is not conditioned on $C$. Hence, we can further derive $P(Y, D \mid C, S) = P(Y \mid C)P(D \mid S)$.

However, the aforementioned spurious correlation frequently appears when facing OOD examples (Arjovsky et al., 2019). As a consequence, when variational inference is based on the factorization in Eq. 1, the approximated content $\tilde{C}$ and style $\tilde{S}$ could both directly influence $Y$ and $D$, i.e., $Y \leftarrow \tilde{C} \rightarrow D$ and $Y \leftarrow \tilde{S} \rightarrow D$, thus leading to inaccurate approximations. However, the desired condition is $Y \leftarrow C \nrightarrow D$ and $Y \nleftarrow S \rightarrow D$. We can see that the unwanted correlations $\tilde{C} \rightarrow D$ and $\tilde{S} \rightarrow Y$ in Fig. 1 (b) is caused by erroneous posteriors $P(D \mid \tilde{C})$ and $P(Y \mid \tilde{S})$. Therefore, to break the correlations, the posteriors $q_{\phi_s}(D \mid C)$ and $q_{\phi_c}(Y \mid S)$ which are correspondingly approximated by the decoders $\phi_s$ and $\phi_c$ can be used as denominators to $q_{\phi_c}(Y \mid C)$ and $q_{\phi_s}(D \mid S)$, respectively. In this way, we can successfully disentangle content $C$ and style $S$ and ensure the decoding process of $Y$ and $D$ would not be influenced by spurious features from $S$ and $C$, respectively. To this end, our factorization in Eq. 1 can be approximated as:

$$\tilde{P}(X, Y, D, C, S) \coloneqq \frac{P(C)P(S)P(Y \mid C)P(D \mid S)P(X \mid C, S)}{q_{\phi_s}(D \mid C)q_{\phi_c}(Y \mid S)}. \tag{2}$$

Then, we maximize the log-likelihood of the joint distribution $p(\mathbf{x}, y, d)$ of each data point $(\mathbf{x}, y, d)$:

$$\log p(\mathbf{x}, y, d) \coloneqq \log \int_c \int_s \tilde{p}(\mathbf{x}, y, d, c, s) \mathrm{d}c \mathrm{d}s, \tag{3}$$

in which we use lower case to denote the values of corresponding variables. Due to the integration of latents $C$ and $S$ is intractable, we follow variational inference (Blei et al., 2017) to obtain an approximated evidence lower-bound $\tilde{ELBO}(\mathbf{x}, y, d)$ of the log-likelihood in Eq. 3:

$$\log p(\mathbf{x}, y, d) \geq \mathbb{E}_{(c,s) \sim q_\theta(C,S|\mathbf{x})} \left[ \log \frac{\tilde{p}(\mathbf{x}, y, d, c, s)}{q_\theta(c, s \mid \mathbf{x})} \right] := \tilde{ELBO}(\mathbf{x}, y, d). \tag{4}$$

Recall the modified joint distribution factorization in Eq. 2, we can have:

$$
\begin{aligned}
\tilde{ELBO}(\mathbf{x}, y, d) =& \mathbb{E}_{(c,s) \sim q_\theta(C,S|\mathbf{x})} \left[ \log \frac{p(c)p(s)q_{\phi_c}(y \mid c)q_{\phi_s}(d \mid s)p_\psi(\mathbf{x} \mid c, s)}{q_\theta(c, s \mid \mathbf{x})q_{\phi_c}(y \mid s)q_{\phi_s}(d \mid c)} \right] \\
=& - KL(q_{\theta_c}(c \mid \mathbf{x}) \| p(C)) - KL(q_{\theta_s}(s \mid \mathbf{x}) \| p(S)) \\
& + \mathbb{E}_{c \sim q_{\theta_c}(C|\mathbf{x})} \left[ \log q_{\phi_c}(y \mid c) - \log q_{\phi_s}(d \mid c) \right] \\
& + \mathbb{E}_{s \sim q_{\theta_s}(S|\mathbf{x})} \left[ \log q_{\phi_s}(d \mid s) - \log q_{\phi_c}(y \mid s) \right] \\
& + \mathbb{E}_{(c,s) \sim q_\theta(C,S|\mathbf{x})} \left[ \log p_\psi(\mathbf{x} \mid c, s) \right] \tag{5a} \\
=& ELBO(\mathbf{x}, y, d) - \mathbb{E}_{c \sim q_{\theta_c}(C|\mathbf{x})} \left[ \log q_{\phi_s}(d \mid c) \right] - \mathbb{E}_{s \sim q_{\theta_s}(S|\mathbf{x})} \left[ \log q_{\phi_c}(y \mid s) \right]. \tag{5b}
\end{aligned}
$$

In Eq. 5a, the first two terms indicate the Kullback-Leibler divergence between the latent variables $C$ and $S$ and their prior distributions. In practice, we assume that the priors $p(C)$ and $p(S)$ follow standard multivariate Gaussian distributions. The third and fourth terms contain the approximated log-likelihoods of label predictions and the disentanglement of the content and style. The last term stands for estimated distribution of $\mathbf{x}$. Note that in Eq. 5b, our approximated $\tilde{ELBO}$ is composed of two parts: the original $ELBO$ which could be obtained from the factorization in Eq. 1, and two regularization terms that aims to disentangle $C$ and $S$ through maximizing the log-likelihoods $\log q_{\phi_s}(d \mid c)$ and $\log q_{\phi_c}(y \mid s)$, which is shown by the dashed lines in Fig. 2. By maximizing $\tilde{ELBO}$, we can train an accurate class predictor which is invariant to different styles. The detailed derivation is provided in supplementary material. Next, we introduce our data augmentation to assist in harnessing OOD examples.

## 2.2 DATA AUGMENTATION WITH CONTENT AND STYLE INTERVENTION

After disentangling content and style, we try to harness OOD examples via two opposite augmentation procedures, namely *positive data augmentation* and *negative data augmentation* which aim to produce benign OOD data $\bar{\mathbf{x}}$ and malign OOD data $\hat{\mathbf{x}}$, respectively, so as to further enhance model generalization and improve robustness against anomalies. Specifically, to achieve this, positive data augmentation only conducts intervention on the style feature meanwhile keeping the content information the same; and the negative data augmentation attempts to affect the content feature while leaving the style unchanged, so as to produce malign OOD data, as shown in Fig. 1 (b).

To achieve this goal, we employ adversarial data augmentation (Goodfellow et al., 2014; Miyato et al., 2018; Volpi et al., 2018) which can directly conduct intervention on the latent variables without influencing each other, thus it is perfect for our intuition of augmenting content and style. Particularly, by adding a learnable perturbation $\mathbf{e}$ to each instance $\mathbf{x}$, we can obtain malign OOD data $\hat{\mathbf{x}}$ and benign OOD data $\bar{\mathbf{x}}$ with augmented content and style, respectively. For each data point $(\mathbf{x}, y, d)$, the perturbation $\mathbf{e}$ can be obtained through minimizing the intervention objective $\mathcal{L}(\cdot)$:

$$\mathbf{e} = \underset{\mathbf{e}; \|\mathbf{e}\|_p < \epsilon}{\arg\min} \mathcal{L}(\mathbf{x} + \mathbf{e}, y, d; \theta_c, \phi_c, \theta_s, \phi_s), \tag{6}$$

where $\epsilon$ denotes the magnitude of the perturbation $\mathbf{e}$ with $\ell_p$-norm. Since our goal of positive and negative data augmentation is completely different, here the intervention objective is designed differently for producing $\bar{\mathbf{x}}$ and $\hat{\mathbf{x}}$. For positive data augmentation, the intervention objective is:

$$\mathcal{L}_{pos} = \mathcal{L}_d(g_c(\mathbf{x}; \theta_c), g_c(\mathbf{x} + \mathbf{e}; \theta_c)) - \mathcal{L}_{ce}(f_s(g_s(\mathbf{x} + \mathbf{e}; \theta_s); \phi_s), d), \tag{7}$$

where the first term $\mathcal{L}_d(\cdot)$ indicates the distance measured between the contents extracted from the original instance and its perturbed version, and the second term $\mathcal{L}_{ce}(\cdot)$ denotes the cross-entropy loss. By minimizing $\mathcal{L}_{pos}$, the perturbation $\mathbf{e}$ would not significantly affect the content feature, meanwhile introducing a novel style that is distinct from its original domain $d$. Consequently, the augmented

benign data with novel styles can be utilized to train a style-invariant model that is resistant to domain shift. Moreover, a specific style with domain label $d'$ can be injected via modifying $\mathcal{L}_{pos}$ as:

$$\mathcal{L}'_{pos} = \mathcal{L}_d(g_c(\mathbf{x}; \theta_c), g_c(\mathbf{x} + \mathbf{e}; \theta_c)) + \mathcal{L}_{ce}(f_s(g_s(\mathbf{x} + \mathbf{e}; \theta_s); \phi_s), d'). \quad (8)$$

Different from Eq. 7, we hope to minimize the cross-entropy loss such that the perturbed instance can contain the style information from a target domain $d'$. As a result, the augmented benign data can successfully bridge the gap between source domain and target domain, and further improve the test performance in the target distribution.

As for negative data augmentation, the intervention objective is defined as:

$$\mathcal{L}_{neg} = \mathcal{L}_d(g_s(\mathbf{x}; \theta_s), g_s(\mathbf{x} + \mathbf{e}; \theta_s)) - \mathcal{L}_{ce}(f_c(g_c(\mathbf{x} + \mathbf{e}; \theta_c); \phi_c), y). \quad (9)$$

By minimizing $\mathcal{L}_{neg}$, the perturbation would not greatly change the style information but would deviate the content from its original one with class label $y$. Subsequently, by recognizing the augmented malign data as unknown, the trained model would be robust to deceiving anomalies with familiar styles, thus boosting the OOD detection performance.

To accomplish the adversarial data augmentation process, here we perform multi-step projected gradient descent (Madry et al., 2018; Wang et al., 2021). Formally, the optimal $\bar{\mathbf{x}}$ and $\hat{\mathbf{x}}$ can be iteratively found through:

$$\bar{\mathbf{x}}^{t+1} = \bar{\mathbf{x}}^t + \operatorname*{arg\,min}_{\mathbf{e}^t; \|\mathbf{e}^t\|_p < \epsilon} \mathcal{L}_{pos}(\bar{\mathbf{x}}^t + \mathbf{e}^t), \hat{\mathbf{x}}^{t+1} = \hat{\mathbf{x}}^t + \operatorname*{arg\,min}_{\mathbf{e}^t; \|\mathbf{e}^t\|_p < \epsilon} \mathcal{L}_{neg}(\hat{\mathbf{x}}^t + \mathbf{e}^t). \quad (10)$$

where the final iteration $t$ is set to 15 in practice. Further, the optimal augmented data will be incorporated into model training, which is described in the next section.

### 2.3 MODEL TRAINING WITH BENIGN AND MALIGN OOD DATA

Finally, based on the aforementioned disentanglement and data augmentation in Sections 2.1 and 2.2, we can obtain a benign OOD data $\bar{\mathbf{x}}$ and a malign OOD data $\hat{\mathbf{x}}$ from each data point $(\mathbf{x}, y, d)$, which will be appended to the benign dataset $\bar{\mathcal{D}}$ and malign dataset $\hat{\mathcal{D}}$, respectively. For utilization of benign OOD data $\bar{\mathbf{x}}$, we assign it with the original class label $y$ and perform supervised training. For separation of malign OOD data $\hat{\mathbf{x}}$, we employ a one-vs-all classifier (Padhy et al., 2020) to recognize them as unknown data that is distinct from its original class label $y$. The proposed HOOD method is summarized in Algorithm 1. Below, we specify the proposed HOOD algorothm to three typical applications with OOD data, namely OOD detection, open-set SSL, and open-set DA.

---

**Algorithm 1** Training process of HOOD

1: Labeled set $\mathcal{D}^l = \{(\mathbf{x}_i, y_i)\}_{i=1}^l$, unlabeled set $\mathcal{D}^u = \{(\mathbf{x}_i)\}_{i=1}^u$.
2: **for** $i = 1$ to $Max\_Iter$ **do**
3:     Pre-train the variational inference framework through maximizing $ELBO$ in Eq. 5a;
4:     Assigning pseudo labels $y^{ps}$ for unlabeled data $\mathcal{D}^u := \{(\mathbf{x}_i; y_i^{ps})\}_{i=1}^u$;
5:     **if** $i == Augmentation\_Iter$ **then**
6:         Conduct Adversarial Data Augmentation to obtain $\bar{\mathbf{x}}$ and $\hat{\mathbf{x}}$ through Eq. 10;
7:         Add $\bar{\mathbf{x}}$ and $\hat{\mathbf{x}}$ into $\bar{\mathcal{D}}$ and $\hat{\mathcal{D}}$, respectively;
8:     **end if**
9:     Enumerate $\bar{\mathcal{D}}$ and conduct supervised training for each $\bar{\mathbf{x}}$;
10:     Enumerate $\hat{\mathcal{D}}$ and recognize each $\hat{\mathbf{x}}$ as unknown;
11: **end for**

---

### 2.4 DEPLOYMENT TO OOD APPLICATIONS

Generally, in all three investigated applications, we are given a labeled set $\mathcal{D}^l = \{(\mathbf{x}_i, y_i)\}_{i=1}^l$ containing $l$ labeled examples drawn from data distribution $P^l$, and an unlabeled set $\mathcal{D}^u = \{\mathbf{x}_i\}_{i=1}^u$ composed of $u$ unlabeled examples sampled from data distribution $P^u$. Moreover, the label space of $\mathcal{D}^l$ and $\mathcal{D}^u$ are defined as $\mathcal{Y}^l$ and $\mathcal{Y}^u$, respectively.

**OOD detection.** The labeled set is used for training, and the unlabeled set is used as a test set which contains both ID data and malign OOD data. Particularly, the data distribution of unlabeled ID data $Q_{id}$ is the same as distribution $P$, but the distribution of OOD data $P^u_{ood}$ is different from $P$, i.e., $P^u_{id} = P^l \neq P^u_{ood}$. The goal is to correctly distinguish OOD data from ID data in the test phase. During training, we conduct data augmentation to obtain domain label $d$, and then follow the workflow described by Algorithm 1 to obtain the model parameters. During test, we only use the

Table 1: Comparison with typical OOD detections methods. Averaged AUROC (%) with standard deviations are computed over three independent trails. The best results are highlighted in **bold**.

| OOD dataset | LSUN | DTD | CUB | Flowers | Caltech | Dogs |
|---|---|---|---|---|---|---|
| ID dataset | | | | SVHN | | |
| Likelihood | $52.25 \pm 0.3$ | $50.33 \pm 0.7$ | $48.76 \pm 0.6$ | $47.33 \pm 0.2$ | $51.54 \pm 0.4$ | $54.34 \pm 0.4$ |
| ODIN | $55.72 \pm 0.2$ | $53.32 \pm 0.5$ | $52.70 \pm 0.4$ | $50.47 \pm 0.7$ | $56.41 \pm 0.4$ | $61.16 \pm 0.3$ |
| Likelihood Ratio | $79.34 \pm 0.5$ | $78.42 \pm 0.3$ | $75.90 \pm 0.7$ | $74.53 \pm 0.4$ | $76.25 \pm 0.3$ | $83.55 \pm 0.4$ |
| OpenGAN | $83.77 \pm 0.4$ | $80.36 \pm 0.5$ | $77.49 \pm 0.8$ | $79.26 \pm 0.5$ | $\mathbf{86.66 \pm 0.5}$ | $86.84 \pm 0.5$ |
| HOOD | $\mathbf{84.10 \pm 0.6}$ | $\mathbf{80.68 \pm 0.6}$ | $\mathbf{79.24 \pm 0.5}$ | $\mathbf{80.93 \pm 0.7}$ | $85.34 \pm 0.7$ | $\mathbf{87.58 \pm 0.8}$ |
| ID dataset | | | | CIFAR10 | | |
| Likelihood | $54.32 \pm 0.5$ | $52.16 \pm 0.4$ | $50.67 \pm 0.4$ | $49.26 \pm 0.3$ | $53.86 \pm 0.4$ | $56.92 \pm 0.2$ |
| ODIN | $58.60 \pm 0.3$ | $55.59 \pm 0.6$ | $58.48 \pm 0.7$ | $51.44 \pm 0.9$ | $59.36 \pm 0.4$ | $64.82 \pm 0.5$ |
| Likelihood Ratio | $81.41 \pm 0.6$ | $79.77 \pm 0.5$ | $79.35 \pm 0.8$ | $77.17 \pm 0.7$ | $80.67 \pm 0.5$ | $86.76 \pm 0.3$ |
| OpenGAN | $84.03 \pm 0.4$ | $81.29 \pm 0.8$ | $82.84 \pm 1.0$ | $\mathbf{82.32 \pm 0.4}$ | $86.78 \pm 0.3$ | $90.14 \pm 0.5$ |
| HOOD | $\mathbf{86.12 \pm 0.6}$ | $\mathbf{83.64 \pm 0.5}$ | $\mathbf{83.53 \pm 0.6}$ | $81.56 \pm 0.8$ | $\mathbf{87.24 \pm 0.8}$ | $\mathbf{90.86 \pm 0.6}$ |

content branch to predict the OOD score which is produced by the one-vs-all classifier. An instance is considered as an ID datum if the OOD score is smaller than $0.5$, and an OOD datum otherwise.

**Open-set SSL.** The labeled set $\mathcal{D}^l$ and unlabeled set $\mathcal{D}^u$ are both used for training, and they are sampled from the same data distribution with different label spaces. Specifically, the unlabeled data contain some ID data that have the same classes as $\mathcal{D}^l$, and the rest unlabeled OOD data are from some unknown classes that do not exist in $\mathcal{D}^l$, formally, $\mathcal{Y}^l \subset \mathcal{Y}^u, \mathcal{Y}^u \setminus \mathcal{Y}^l \neq \varnothing$ and $P^l(\mathbf{x} \mid y) = P^u(\mathbf{x} \mid y), y \in \mathcal{Y}^l$. The goal is to properly leverage the labeled data and unlabeled ID data without being misled by malign OOD data, and correctly classify test data with labels in $\mathcal{Y}^l$. The training process is similar to OOD detection, except that HOOD would produce an OOD score for each unlabeled data. If an unlabeled instance is recognized as OOD data, it would be left out.

**Open-set DA.** The labeled set is drawn from source distribution $P^l$ which is different from the target distribution $P^u$ of unlabeled set. In addition, the label space $\mathcal{Y}^l$ is also a subset of $\mathcal{Y}^u$. Therefore, the unlabeled data consist of benign OOD data which have the same class labels as labeled data, and malign OOD data which have distinct data distribution as well as class labels from labeled data, formally, $P^l \neq P^u, \mathcal{Y}^l \subset \mathcal{Y}^u, \mathcal{Y}^u \setminus \mathcal{Y}^l \neq \varnothing$. The goal is to transfer the knowledge of labeled data to the benign OOD data, meanwhile identify the malign OOD data as unknown. In this application, we assign each target instance with a domain label to distinguish them from other augmented data. Then we alter the positive data augmentation objective from Eq. 7 to Eq. 8 and train the framework through Algorithm 1. During test, HOOD would predict each target instance as some class if it is benign OOD data, and as unknown otherwise.

## 3 EXPERIMENT

In this section, we first describe the implementation details. Then, we experimentally validate our method on three applications, namely OOD detection, open-set SSL, and open-set DA. Finally, we present extensive performance analysis on our disentanglement and intervention modules. Additional details and quantitative findings can be found in the supplementary material.

### 3.1 IMPLEMENTATION DETAILS

In experiments, we choose Wide ResNet-28-2 (Zagoruyko & Komodakis, 2016) for OOD detection and Open-set SSL tasks, and follow (You et al., 2020; Cao et al., 2019) to utilize ResNet50 pre-trained on Imagenet (Russakovsky et al., 2015) for Open-set DA. For implementing HOOD, we randomly choose 4 augmentation methods from the transformation pool in RandAugment (Cubuk et al., 2020), to simulate different styles. The pre-training iteration $Augmentation\_Iter$ is set to 100,000, and the perturbation magnitude $\epsilon = 0.03$, following (Volpi et al., 2018) in all experiments. Next, we will experimentally validate HOOD in three applications.

### 3.2 OOD DETECTION

In OOD detection task, we use SVHN (Netzer et al., 2011) and CIFAR10 (Krizhevsky et al., 2009) as the ID datasets, and use LSUN (Yu et al., 2015), DTD (Cimpoi et al., 2014), CUB (Wah et al., 2011), Flowers (Nilsback & Zisserman, 2006), Caltech (Griffin et al., 2007), and Dogs (Khosla et al., 2011) datasets as the OOD datasets that occur during test phase. Particularly, to explore the model generalization ability, we only sample 100 labeled data and 20,000 unlabeled data from each class and conduct semi-supervised training, then we test the trained model on the unlabeled OOD dataset.

Table 2: Comparison with typical Open-set SSL methods. Averaged test accuracies (%) with standard deviations are computed over three independent trails. The best results are highlighted in **bold**.

| Training dataset | | CIFAR10 | | | CIFAR100 | | |
|---|---|---|---|---|---|---|---|
| No. of Labeled data | | 50 | 100 | 400 | 50 | 100 | 400 |
| Clean Acc. | UASD | $72.82 \pm 0.9$ | $75.53 \pm 1.8$ | $76.74 \pm 1.7$ | $58.87 \pm 0.6$ | $61.68 \pm 1.2$ | $65.97 \pm 2.4$ |
| | DS3L | $74.44 \pm 1.3$ | $76.89 \pm 1.5$ | $78.80 \pm 0.6$ | $60.40 \pm 0.5$ | $64.35 \pm 1.5$ | $67.65 \pm 1.3$ |
| | MTCF | $79.88 \pm 1.3$ | $81.41 \pm 1.0$ | $83.92 \pm 0.8$ | $62.78 \pm 0.5$ | $65.84 \pm 2.1$ | $69.46 \pm 0.6$ |
| | OpenMatch | $\mathbf{84.10 \pm 1.1}$ | $\mathbf{85.30 \pm 0.4}$ | $\mathbf{87.92 \pm 1.0}$ | $65.76 \pm 0.9$ | $\mathbf{68.46 \pm 0.5}$ | $72.87 \pm 1.4$ |
| | T2T | $82.74 \pm 1.2$ | $83.56 \pm 1.4$ | $85.97 \pm 0.8$ | $65.16 \pm 1.2$ | $67.58 \pm 0.9$ | $71.96 \pm 1.1$ |
| | HOOD | $83.55 \pm 1.2$ | $84.16 \pm 1.5$ | $86.22 \pm 2.7$ | $\mathbf{66.39 \pm 1.7}$ | $68.03 \pm 2.6$ | $\mathbf{73.32 \pm 0.6}$ |
| Corrupted Acc. | UASD | $39.36 \pm 1.2$ | $41.38 \pm 0.7$ | $42.66 \pm 1.8$ | $31.55 \pm 2.0$ | $33.39 \pm 1.7$ | $35.20 \pm 0.8$ |
| | DS3L | $39.97 \pm 0.8$ | $42.58 \pm 0.8$ | $44.39 \pm 0.6$ | $33.72 \pm 0.8$ | $34.67 \pm 0.8$ | $36.64 \pm 0.6$ |
| | MTCF | $40.16 \pm 1.2$ | $40.58 \pm 1.1$ | $43.33 \pm 0.7$ | $32.72 \pm 0.8$ | $34.33 \pm 2.3$ | $35.53 \pm 0.6$ |
| | OpenMatch | $41.38 \pm 0.7$ | $42.90 \pm 0.6$ | $45.79 \pm 0.8$ | $35.98 \pm 1.3$ | $36.47 \pm 0.7$ | $38.56 \pm 0.6$ |
| | T2T | $41.39 \pm 1.6$ | $45.56 \pm 1.6$ | $49.88 \pm 1.5$ | $\mathbf{41.03 \pm 1.7}$ | $39.64 \pm 0.7$ | $41.38 \pm 1.6$ |
| | HOOD | $\mathbf{44.42 \pm 1.7}$ | $\mathbf{48.38 \pm 0.9}$ | $\mathbf{50.74 \pm 0.6}$ | $40.82 \pm 1.5$ | $\mathbf{41.65 \pm 0.9}$ | $\mathbf{43.72 \pm 2.2}$ |

To evaluate the performance, we utilize AUROC (Hendrycks & Gimpel, 2017) which is an essential metric for OOD detection, and a higher AUROC value indicates a better performance.

For comparison, we choose some typical OOD detection methods including Likelihood (Hendrycks & Gimpel, 2017) which simply utilizes softmax score as the detection criterion, ODIN (Liang et al., 2018) which enhances the performance of Likelihood through adding adversarial attack, Likelihood Ratio (Ren et al., 2019) which modifies the softmax score through focusing on the semantic feature, and OpenGAN (Kong & Ramanan, 2021) which can further improve the performance via separating the generated "hard" examples that are deceivingly close to ID data.

The experimental results are shown in Table 1, we can see that HOOD can greatly surpass Likelihood, ODIN, and Likelihood Ratio, and can outperform OpenGAN in most scenarios. When compared with softmax-prediction-based methods such as Likelihood and ODIN, HOOD surpasses them in a large margin, as HOOD can correctly separate some overconfident OOD examples from ID data. As for Likelihood Ratio, our method can achieve better performance through producing "hard" malign OOD data, thus successfully avoiding deceiving examples that are extremely close to ID data. Although both OpenGAN and HOOD generate "hard" malign data to train an open classifier, HOOD can successfully distinguish content and style thanks to the aforementioned disentanglement, thus avoid rejecting too much benign OOD data and further yield better detection performance than OpenGAN.

### 3.3 OPEN-SET SSL

In open-set SSL task, we follow (Guo et al., 2020) to construct our training dataset using two benchmark datasets CIFAR10 and CIFAR100 (Krizhevsky et al., 2009), which contains 10 and 100 classes, respectively. The constructed dataset has 20,000 randomly sampled unlabeled data and a varied number of labeled data. Here the number of labeled data is set to 50, 100, and 400 per class in both CIFAR10 and CIFAR100. Moreover, to create the open-set problem in CIFAR10, the unlabeled data is sampled from all 10 classes and the labeled data is sampled from the 6 animal classes. As for CIFAR100, the unlabeled data are sampled from all 100 classes and the labeled data is sampled from the first 60 classes. For evaluation, we first use the test dataset from the original CIFAR10 and CIFAR100 and denote the test accuracy as "Clean Acc.". Further, to evaluate the capability of handling OOD examples, we test on CIFAR10-C and CIFAR100-C (Hendrycks & Dietterich, 2019) which add different types of corruptions to CIFAR10 and CIFAR100, respectively. The test accuracy from the corrupted datasets can reveal the robustness of neural networks against corruptions and perturbations, and it is denoted as "Corrupted Acc.".

For comparison, we choose some typical open-set SSL methods including Uncertainty-Aware Self-Distillation method UASD (Chen et al., 2020) and T2T (Huang et al., 2021a) which filters out the OOD data via using OOD detection, Safe Deep Semi-Supervised Learning DS3L (Guo et al., 2020) which employs meta-learning to down-weight the OOD data, Multi-Task Curriculum Framework MTCF (Yu et al., 2020) which recognizes the OOD data as different domain, and OpenMatch (Saito et al., 2021) which utilizes open-set consistency training on OOD data.

The experimental results are shown in Table 2. Compared to the strongest baseline method Open-Match, which randomly samples eleven different transformations from a transformation pool, our method has transformations that are limited to only four types. In CIFAR10 and CIFAR100 regarding the Clean Acc., the proposed HOOD is slightly outperformed by OpenMatch. However, thanks to

Table 3: Comparison with typical Open-set DA methods. Averaged test accuracies (%) with standard deviations are computed over three independent trails. The best results are highlighted in **bold**.

| Dataset | Office | | | | | | VisDA |
|---|---|---|---|---|---|---|---|
| Domain | A→W | A→D | D→W | W→D | D→A | W→A | Synthetic→Real |
| OSBP | $86.5 \pm 2.0$ | $88.6 \pm 1.4$ | $97.0 \pm 1.0$ | $97.9 \pm 0.9$ | $88.9 \pm 2.5$ | $85.8 \pm 2.5$ | $62.9 \pm 1.3$ |
| UAN | $87.7 \pm 1.2$ | $87.0 \pm 0.8$ | $93.5 \pm 1.3$ | $97.2 \pm 1.6$ | $88.4 \pm 0.7$ | $87.8 \pm 1.6$ | $63.8 \pm 2.4$ |
| STA | $89.5 \pm 0.6$ | $93.7 \pm 1.5$ | $97.5 \pm 0.2$ | $\mathbf{99.5 \pm 0.2}$ | $89.1 \pm 0.5$ | $87.9 \pm 0.9$ | $66.4 \pm 1.3$ |
| HOOD | $\mathbf{90.1 \pm 1.5}$ | $\mathbf{94.2 \pm 1.4}$ | $\mathbf{99.6 \pm 0.6}$ | $98.3 \pm 0.9$ | $\mathbf{89.8 \pm 0.8}$ | $\mathbf{91.3 \pm 1.8}$ | $\mathbf{72.4 \pm 1.6}$ |

the disentanglement, HOOD can be invariant to different styles and focus on the content feature. Therefore, when facing corruption, HOOD can be more robust than all baseline methods. As shown by the Corrupted Acc. results, our method surpasses OpenMatch for more than 3%.

## 3.4 OPEN-SET DA

In open-set DA task, we follow (Saito et al., 2018) to validate on two DA benchmark datasets Office (Saenko et al., 2010) and VisDA (Peng et al., 2018). Office dataset contains three domains Amazon (A), Webcam (W), and DSLR (D), and each domain is composed of 31 classes. VisDA dataset contains two domains Sythetic and Real, and each domain consists of 12 classes. To create an open-set situation in Office, we follow (Saito et al., 2018; Liu et al., 2019) to construct the source dataset by sampling from the first 21 classes in alphabetical order. Then, the target dataset is sampled from all 31 classes. As for VisDA, we choose the first 6 classes for source domain, and use all the 12 classes for target domain. We use "A→W" to indicate the transfer from "A" domain to "W" domain.

For comparison, we choose three typical open-set DA approaches including Open-Set DA by Back-Propagation OSBP (Saito et al., 2018) which employs an OpenMax classifier to recognize unknown classes and perform gradient flipping for open-set DA, Universal Adaptation Network UAN (You et al., 2020) which utilize entropy and domain similarity to down-weight malign OOD data, and Separate To Adapt STA (Liu et al., 2019) which utilizes SVM to separate the malign OOD data.

The experimental results are shown in Table 3. Compared to the baseline methods, the proposed HOOD is largely benefited from the generated benign OOD data, which have two major strengths: 1) they resemble target domain data by having common styles, and 2) their labels are accessible as they share the same content as their corresponding source data. Therefore, through conducting supervised training such benign OOD data, the domain gap can be further mitigated, thus achieving better performance than baseline methods. Quantitative results show that HOOD can surpass other methods in most scenarios. Especially in VisDA, HOOD can outperform the second-best method with 6% improvement, which proves the effectiveness of HOOD in dealing with open-set DA.

## 3.5 PERFORMANCE ANALYSIS

**Ablation Study:** To verify the effectiveness of each module, we conduct an ablation study on three OOD applications by eliminating one component at a time. Specifically, our HOOD can be ablated into: "w/o disentanglement" which indicates removing the

Table 4: Ablation study on necessity of each module.

| Application | OOD detection | Open-Set SSL | Open-Set DA |
|---|---|---|---|
| w/o disentanglement | $84.94 \pm 1.3$ | $82.55 \pm 1.1$ | $64.6 \pm 0.9$ |
| w/o benign OOD data | $85.95 \pm 1.8$ | $83.32 \pm 2.0$ | $66.3 \pm 2.5$ |
| w/o malign OOD data | $82.50 \pm 2.2$ | $85.40 \pm 0.8$ | $71.8 \pm 1.2$ |
| w/o both augmentations | $80.83 \pm 0.8$ | $81.14 \pm 1.2$ | $65.4 \pm 1.2$ |
| HOOD | $\mathbf{86.12 \pm 0.6}$ | $\mathbf{86.22 \pm 2.7}$ | $\mathbf{72.4 \pm 1.6}$ |

disentanglement loss in Eq. 5a, "w/o benign OOD data" which denotes training without benign OOD data, "w/o malign OOD data" which stands for discarding malign OOD data, and "w/o both augmentations" indicates training without both benign and malign OOD data. Here in OOD detection, we use CIFAR10 as the ID dataset, and use LSUN as the OOD dataset. In open-set SSL, we choose CIFAR10 with 400 labels for each class. As for open-set DA, we use VisDA dataset.

The experimental results are shown in Table 4. We can see each module influences the performance differently in three applications. First, we can see that the malign OOD data is essential for OOD detection, as it can act as unknown anomalies and reduce the overconfidence in unseen data. Then, benign OOD data can largely improve the learning performance in open-set SSL and open-set DA, as they can enforce the model to focus on the content feature for classification. Additionally, we can see that discarding both benign and malign OOD data shows performance degradation compared to both "w/o benign OOD data" and "w/o malign OOD data". Therefore, our HOOD can correctly change the style and content, which can correspondingly benefit generalization tasks (such as open-set DA

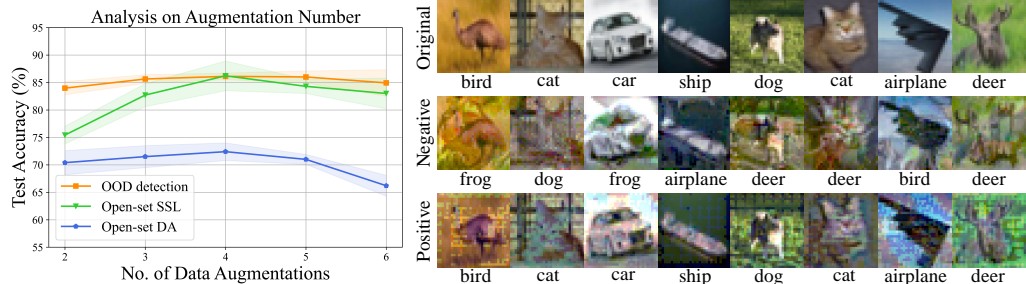

Figure 4: Augmentation number analysis.    Figure 5: CIFAR10 Visualization of our data augmentation.

and open-set SSL) and detection tasks (such as OOD detection). Moreover, open-set DA relies more on the disentanglement than the rest two modules, owing to the disentanglement can exclude the style changing across different domains. Hence, our disentanglement can effectively eliminate the distribution change from different domains and help learn invariant features.

**Analysis on Augmentation Number:** Since HOOD does not introduce any hyper-parameter, the most influential setting is the number of data augmentation. To analyze its influence on the learning results, we vary the number of augmentations that are sampled from the RandAugment Pool (Cubuk et al., 2020) from 2 to 6. The results are shown in Fig. 4. We can see that both too less and too many augmentations would hurt the results. This is because a small augmentation number would undermine the generalization to various styles; and a large augmentation number would increase the classification difficulty of the style branch, further making the disentanglement hard to achieve. Therefore, setting the augmentation number to 4 is reasonable.

**Visualization:** Furthermore, to show the effect of our data augmentations, we visualize the augmented images by applying large perturbation magnitude (4.7) Tsipras et al. (2018) in Fig. 5. The model prediction is shown below each image. We can see that the negative data augmentation significantly changes the content which is almost unidentifiable. However, positive data augmentation can still preserve most of the content information and only change the style of images. Therefore, the augmented data can be correctly leveraged to help train a robust classifier.

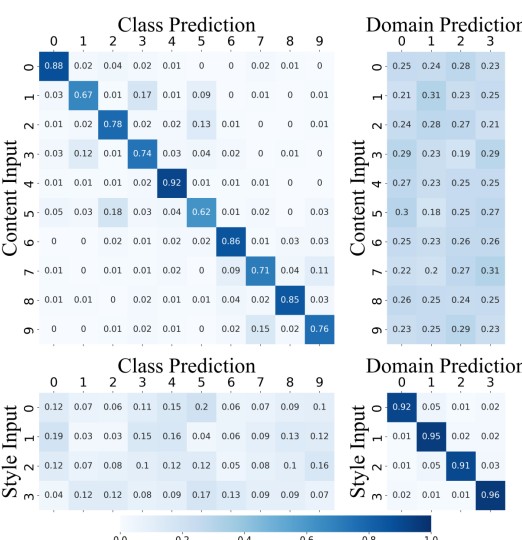

Figure 3: Illustration of disentanglement between content and style on CIFAR10. The number in each cell denotes the prediction probability.

**Disentanglement of Content and Style:** To further testify that our disentanglement between content and style is effective, we select the latent variables from different content and style categories, and use the learned class and domain classifiers for cross-prediction. Specifically, there are four kinds of input-prediction types: content-class, content-domain, style-class, and style-domain. As we can see in Fig. 3, only the content features are meaningful for class prediction, and the same phenomenon goes for style input and domain prediction. However, neither of the style and content features can be identified by the class predictor and domain predictor, respectively. Therefore, we can reasonably conclude that our disentanglement between content and style is effectively achieved.

## 4    CONCLUSION

In this paper, we propose HOOD to effectively harness OOD examples. Specifically, we construct a SCM to disentangle content and style, which can be leveraged to identify benign and malign OOD data. Subsequently, by maximizing ELBO, we can successfully disentangle the content and style feature and break the spurious correlation between class and domain. As a result, HOOD can be more robust when facing distribution shifts and unseen OOD data. Furthermore, we augment the content and style through a novel intervention process to produce benign and malign OOD data, which can be leveraged to improve classification and OOD detection performance. Extensive experiments are conducted to empirically validate the effectiveness of HOOD on three typical OOD applications.

## 5 ACKNOWLEDGEMENT

Li Shen was partially supported by the Major Science and Technology Innovation 2030 "New Generation Artificial Intelligence" key project No. 2021ZD0111700. Zhuo Huang was supported by JD Technology Scholarship for Postgraduate Research in Artificial Intelligence No. SC4103. Xiaobo Xia was supported by Australian Research Council Projects DE-190101473 and Google PhD Fellowship. Bo Han was supported by NSFC Young Scientists Fund No. 62006202, Guangdong Basic and Applied Basic Research Foundation No. 2022A1515011652 and RGC Early Career Scheme No. 22200720. Mingming Gong was supported by ARC DE210101624. Chen Gong was supported by NSF of China No. 61973162, NSF of Jiangsu Province No. BZ2021013, NSF for Distinguished Young Scholar of Jiangsu Province No. BK20220080, and CAAI-Huawei MindSpore Open Fund. Tongliang Liu was partially supported by Australian Research Council Projects IC-190100031, LP-220100527, DP-220102121, and FT-220100318.

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

## A  APPENDIX

In this supplementary material, we first discuss some related work in Section B. Then, we complement the implementation details of HOOD in Section C. Further, we present the detailed derivation of $ELBO$ in Section D. Additionally, we discuss two notable causal graph assumptions in Section E. Moreover, to further understand the effectiveness of HOOD, we provide additional analysis in Section F. Finally, we will discuss the limitations and social impact of our method in Section G.

## B  RELATED WORK

**OOD applications** contains three typical problems, namely OOD detection, open-set SSL, and open-set DA. OOD detection Hendrycks & Gimpel (2017); Liang et al. (2018) aims to train a robust model which can accurately identify the newly-emerged malign OOD data during the test phase. Open-set SSL Guo et al. (2020); Chen et al. (2020); He et al. (2022a;b); Yu et al. (2020) deals with the problem when labeled data are scarce and the unlabeled data are contaminated by malign OOD data. As for open-set DA Saito et al. (2018); Liu et al. (2019), it tries to transfer the knowledge from source ID data to the benign OOD data in target domain, meanwhile detecting the malign OOD data that are encountered during transferring. In both three applications, the predictive confidence has been frequently leveraged to separate malign OOD data (Hendrycks & Gimpel, 2017; Liang et al., 2018; Ren et al., 2019; Xia et al., 2022a). Moreover, ID data and OOD data can be distinguished via using a discriminator (Kong & Ramanan, 2021; Neal et al., 2018; Yu et al., 2020; Xia et al., 2021). Further, various open classifiers are designed to predict OOD dataset as unknown (Ge et al., 2017; Padhy et al., 2020; Saito et al., 2018). Thanks to the advances in unsupervised learning, many approaches employ self-supervised learning to make ID data and OOD data separable (Cao et al., 2022; Li et al., 2021a; Saito et al., 2020) from each other.

**Causality in OOD problems** mainly focuses on learning invariant representations that stay constant when other causal factors are changing, thus achieving better performance when facing non-stationary data distribution. To accomplish this goal, it is common to learn causal factors and non-causal factors through the variational auto-encoder framework (Blei et al., 2017; Kingma & Welling, 2013). Thanks to which, domain adaptation (Gong et al., 2016; Schölkopf et al., 2011; Zhang et al., 2013) and domain generalization (Li et al., 2018; Shankar et al., 2018) can be tackled through extracting the domain invariant features. Moreover, based on causal effects, the biased feature can be eliminated through re-weighting (Bahadori et al., 2017; Shen et al., 2018). Additionally, the spurious correlation which is harmful for inference could be alleviated through do-calculus (Lee et al., 2021; Nam et al., 2020; Pearl, 2009). Recent methods (Ilse et al., 2021; Mitrovic et al., 2020; Von Kügelgen et al., 2021) conduct data augmentations with self-supervised learning to train a robust model that can handle distribution shifts and corruptions.

In general, HOOD has two major differences from existing methods in OOD applications and causality. On one hand, instead of treating an image instance as a whole as commonly done in many approaches, HOOD can properly leverage OOD examples through their disentangled contents and styles. Moreover, augmenting content and style can help improve generalization and robustness simultaneously. On the other hand, current causal approaches are incapable of dealing with malign OOD data, but HOOD is able to learn style-invariant features from benign OOD data, meanwhile avoiding the damage brought by malign OOD data.

## C  COMPLEMENTARY DETAILS

Each trial of our experiments is conducted in one single NVIDIA 3090 GPU. In the open-set SSL and OOD detection tasks, we follow (Saito et al., 2021) by using the Wide ResNet-28-2 (Zagoruyko & Komodakis, 2016) as the network backbone and train from scratch. In open-set DA application, we follow (Saito et al., 2018) to fine-tune ResNet50 pre-trained on Imagenet (Russakovsky et al., 2015). The employed Stochastic Gradient Descent (SGD) optimizer starts with an initial learning rate $3e - 2$ which is decayed by following the cosine function $cos(const \times \frac{current\_iteration}{500,000})$ without warm-up, in which $const$ is constant, we follow (Saito et al., 2021) to set it as $\frac{7}{16}\pi$. The momentum factor is set to $0.9$, which is also the same as (Saito et al., 2021). For choosing the pseudo labels of unlabeled data, we follow (Lee, 2013) to set the pseudo label threshold as $0.95$. The unlabeled examples with confidence smaller than $0.95$ are excluded from training the class branch. However, all of them are leveraged to optimize the domain branch. Moreover, the content and style features are the output of

the penultimate layer, which are further fed into the fully-connected layer to produce the class and domain prediction. Here the class number is decided by the class labels, and the domain number is the number of augmentations plus one logit that stands for the original input.

## D DERIVATION OF ELBO

In the main paper, the modified structural causal model is factorized as:

$$P'(X, Y, D, C, S) = \frac{P(C)P(S)P(Y \mid C)P(D \mid S)P(X \mid C, S)}{P(D \mid C)P(Y \mid S)}. \tag{11}$$

We employ two encoders to model the distribution of content and style, respectively:

$$q_{\theta_c}(C \mid X), \; q_{\theta_s}(S \mid X). \tag{12}$$

Moreover, we utilize two classifiers to model the posteriors of class and domain, respectively:

$$q_{\phi_c}(Y \mid C), \; q_{\phi_s}(D \mid S). \tag{13}$$

Additionally, a decoder is employed to reconstruct the input instance through following distribution:

$$q_{\phi_s}(X \mid C, S). \tag{14}$$

Then, our goal is to maximize the log-likelihood of the joint distribution $p(\mathbf{x}, y, d)$:

$$
\begin{aligned}
\log p(\mathbf{x}, y, d) &= \log \int_c \int_s p'(\mathbf{x}, y, d, c, s) \mathrm{d}c \mathrm{d}s \\
&= \log \int_c \int_s p'(\mathbf{x}, y, d, c, s) \frac{q_\theta(c, s \mid \mathbf{x})}{q_\theta(c, s \mid \mathbf{x})} \mathrm{d}c \mathrm{d}s \\
&= \log \mathbb{E}_{(c,s) \sim q_\theta(C,S|\mathbf{x})} \left[ \frac{p'(\mathbf{x}, y, d, c, s)}{q_\theta(c, s \mid \mathbf{x})} \right] \\
&\geq \mathbb{E}_{(c,s) \sim q_\theta(C,S|\mathbf{x})} \left[ \log \frac{p'(\mathbf{x}, y, d, c, s)}{q_\theta(c, s \mid \mathbf{x})} \right] := ELBO(\mathbf{x}, y, d).
\end{aligned} \tag{15}
$$

By applying Eq. 2 to $p'(\mathbf{x}, y, d, c, s)$, we have:

$$
\begin{aligned}
ELBO(\mathbf{x}, y, d) &= \mathbb{E}_{(c,s) \sim q_\theta(C,S|\mathbf{x})} \left[ \log \frac{p(c)p(s)q_{\phi_c}(y \mid c)q_{\phi_s}(d \mid s)p_\psi(\mathbf{x} \mid c, s)}{q_\theta(c, s \mid \mathbf{x})q_{\phi_c}(y \mid s)q_{\phi_s}(d \mid c)} \right] \\
&= \mathbb{E}_{(c,s) \sim q_\theta(C,S|\mathbf{x})} \left[ \log \frac{p(c)p(s)}{q_{\theta_c}(c \mid \mathbf{x})q_{\theta_s}(s \mid \mathbf{x})} \right] + \mathbb{E}_{(c,s) \sim q_\theta(C,S|\mathbf{x})} \left[ \log \frac{q_{\phi_c}(y \mid c)}{q_{\phi_s}(d \mid c)} \right] \\
&\quad + \mathbb{E}_{(c,s) \sim q_\theta(C,S|\mathbf{x})} \left[ \log \frac{q_{\phi_s}(d \mid s)}{q_{\phi_c}(y \mid s)} \right] + \mathbb{E}_{(c,s) \sim q_\theta(C,S|\mathbf{x})} \left[ \log p_\psi(\mathbf{x} \mid c, s) \right] \\
&= \mathbb{E}_{(c) \sim q_{\theta_c}(C|\mathbf{x})} \left[ \log \frac{p(c)}{q_{\theta_c}(c \mid \mathbf{x})} \right] + \mathbb{E}_{(s) \sim q_{\theta_s}(S|\mathbf{x})} \left[ \log \frac{p(c)p(s)}{q_{\theta_s}(s \mid \mathbf{x})} \right] \\
&\quad + \mathbb{E}_{(c) \sim q_{\theta_s}(C|\mathbf{x})} \left[ \log \frac{q_{\phi_c}(y \mid c)}{q_{\phi_s}(d \mid c)} \right] + \mathbb{E}_{(s) \sim q_{\theta_s}(S|\mathbf{x})} \left[ \log \frac{q_{\phi_s}(d \mid s)}{q_{\phi_c}(y \mid s)} \right] \\
&\quad + \mathbb{E}_{(c,s) \sim q_\theta(C,S|\mathbf{x})} \left[ \log p_\psi(\mathbf{x} \mid c, s) \right] \\
&= -KL(q_{\theta_c}(c \mid \mathbf{x}) \| p(C)) - KL(q_{\theta_s}(s \mid \mathbf{x}) \| p(S)) \\
&\quad + \mathbb{E}_{c \sim q_{\theta_c}(C|\mathbf{x})} \left[ \log q_{\phi_c}(y \mid c) - \log q_{\phi_s}(d \mid c) \right] \\
&\quad + \mathbb{E}_{s \sim q_{\theta_s}(S|\mathbf{x})} \left[ \log q_{\phi_s}(d \mid s) - \log q_{\phi_c}(y \mid s) \right] \\
&\quad + \mathbb{E}_{(c,s) \sim q_\theta(C,S|\mathbf{x})} \left[ \log p_\psi(\mathbf{x} \mid c, s) \right],
\end{aligned} \tag{16}
$$

which is the final $ELBO$ in our main paper.

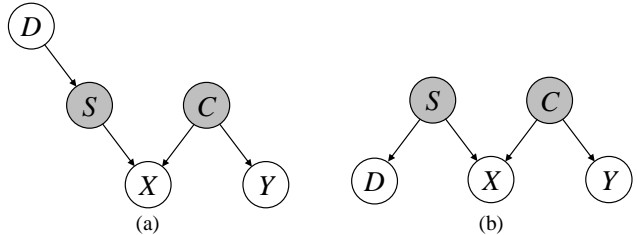

Figure 6: Comparison of two different causal relationship assumptions.

# E    DISCUSSION OF THE CAUSAL RELATIONSHIP BETWEEN STYLE AND DOMAIN

There are two different assumptions for the causal relationship between style $S$ and domain $D$: a) Style is caused by domain; and b) Style causes domain (as shown in Figure 6). The first one is more commonly used in existing literature Liu et al. (2021); Lu et al. (2021); Suter et al. (2019), which usually assume that the style of data depends on the sampling environment. Such an assumption perfectly fits the natural data generating process. However, we follow Mitrovic et al. (2020); Von Kügelgen et al. (2021) by using data augmentation to obtain different styles. During this process, each data augmentation only introduces a specific style, then we manually assign a domain label based on the style type. Hence, in this scenario, assuming that the style causes the domain label as scenario b) is more appropriate than the first one.

Furthermore, the causal direction between style and domain depends on the problem context. If the domain is partitioned according to the style in the images, content is the cause of domain. For example, in image recognition, one can partition images into domains according to the illumination levels. The domain is determined by the brightness of the images and the causal relation is brightness→domain. On the other hand, if one partition the domains according to time (daytime or night), the brightness level will be an effect of the domain (or time), *i.e.*, domain→brightness. Therefore, our assumption b) holds in certain real-world scenarios.

# F    ADDITIONAL ANALYSIS

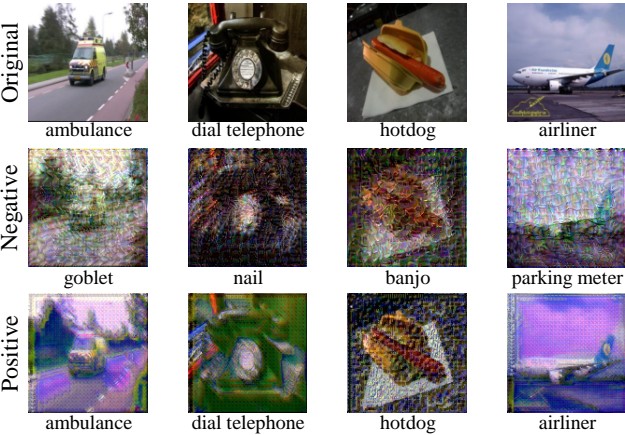

Figure 7: ImageNet30 visualization of our data augmentation. The model prediction is shown below each image.

## F.1    VISUALIZATION OF HIGHER RESOLUTION IMAGES

In the main paper, we have provided the visualization of augmented CIFAR10 images. To testify that our positive data augmentation and negative data augmentation are still effective on higher resolution images, we conduct experiments using ImageNet30 with resolution $256 \times 256$, and show the augmented benign data and malign data in Fig. 7. We can see the similar phenomenon as in

CIFAR10: In negative data augmentation, the objects are completely unidentifiable which lead to erroneous model predictions. On the contrary, the positive data augmentation only changes the style (there are three style types are presented: purple style, green style, and sharp-texture style) but leave the objects intact. As a result, the model predictions are usually correct. Therefore, our augmentation method can be effectively deployed to high resolution images.

## F.2 t-SNE VISUALIZATION

To further demonstrate the effectiveness of our disentanglement, we show the t-SNE (Maaten & Hinton, 2008) visualization of the feature extracted with or without disentanglement, as shown in Fig. 8. We can see that when training without disentanglement, the features are closely gathered in each cluster. However, the malign OOD data represented by gray and pink points are also intensely aligned in the clusters, which would damage the model robustness. In contrast, when training with disentanglement, there is a slight gap between OOD data and ID data, which means that the model trained with disentanglement can avoid overfitting to specific styles, and show better robustness against OOD data.

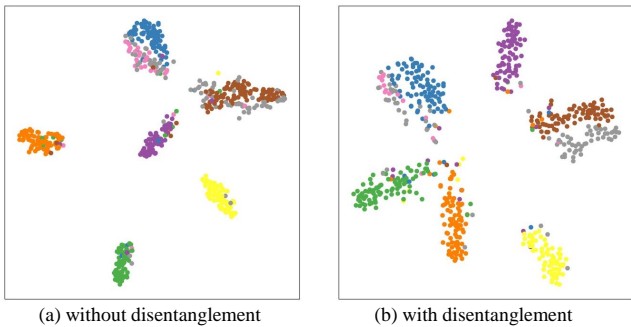

(a) without disentanglement          (b) with disentanglement

Figure 8: t-SNE visualization on CIFAR10 dataset. The blue, brown, yellow, green, orange, and purple points are ID data, and the gray and pink points are OOD data.

## F.3 EFFECT OF ADDING BENIGN AND MALIGN OOD DATA INTO TRAINING

To give a illustrative comparison of adding benign OOD data and malign OOD data into training, we conduct experiments under the open-set SSL setting and separately augmenting benign OOD data and malign OOD data to compare their effects. Moreover, we conduct plain training as a baseline result which do not use either augmentations. The results are shown in Fig. 9. We can see that after adding augmented data, the effect of malign OOD data causes sudden performance degradation. On the contrary, benign OOD data can further improve the learning result compared to the plain training baseline. Which again shows that preserving content and augmenting style is beneficial for improving generalization, and eliminating content is harmful for learning.

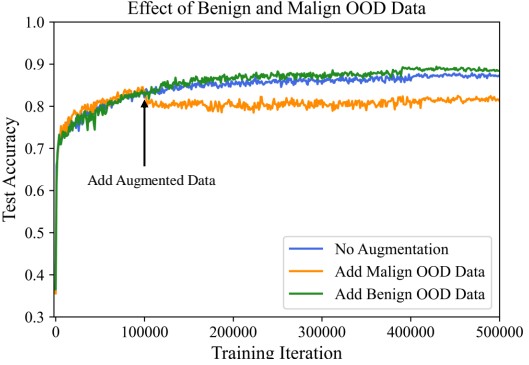

Figure 9: Effect of adding benign and malign OOD data into training.

### F.4 OOD Score

To show the effectiveness of identifying malign OOD data from benign OOD data, we test the performance of HOOD on three applications to observe the OOD scores of benign OOD data and malign OOD data and show the averaged OOD scores in Table 5. We can see that the OOD score produced by our one-vs-all classifier can clearly distinguish benign and malign OOD data during the test phase, which again validates the effectiveness of HOOD.

Table 5: Averaged OOD scores with standard deviations on three applications.

| Application | OOD score | |
|---|---|---|
| | Benign OOD data | Malign OOD data |
| OOD detection | $0.16 \pm 0.3$ | $0.83 \pm 0.6$ |
| Open-Set SSL | $0.08 \pm 0.5$ | $0.91 \pm 0.4$ |
| Open-Set DA | $0.21 \pm 0.4$ | $0.88 \pm 0.3$ |

### F.5 Execution Efficiency

Additionally, to give a quantitative comparison on the execution efficiency of HOOD, here we provide the running time on 3090 GPU compared to some typical baseline methods. The results are shown in Table 6. Note that our method involves causal disentanglement as well as adversarial training, therefore, the training time is more than other opponents.

Table 6: Execution efficiency comparisons on three applications.

| OOD Detection | | Open-set SSL | | Open-set DA | |
|---|---|---|---|---|---|
| Method | Time | Method | Time | Method | Time |
| Likelihood | 6.2h | DS3L | 15.4h | UAN | 8.5h |
| OpenGAN | 7.8h | OpenMatch | 10.5h | STA | 9.1h |
| HOOD | 11.4h | HOOD | 13.7h | HOOD | 12.4h |

## G  Limitation and Social Impact

The proposed HOOD method has many advantages which have been demonstrated in the main paper. However, HOOD is still limited in some aspects. Firstly, HOOD contains an extra phase that computes the gradient to produce the augmented data, so it cannot be conducted in an end-to-end manner. Therefore, it is feasible to improve HOOD by designing a more compact method and incorporating augmentation into the training process. Secondly, HOOD utilizes existing data augmentation techniques to simulate different styles, which cannot perfectly cover all the possible styles in the real world. Hence, better style simulation might further improve the learning performance of HOOD.

Regardless of the limitations, our method could have some positive social impacts. First, HOOD can be safely deployed into many open situations where many unknown classes exist. As practical problems contain lots of uncertainties and novel instances occur constantly. Thanks to the negative data augmentation of HOOD, the novel instances can be successfully identified and would not harm the prediction accuracy. Secondly, in many non-stationary environments, the knowledge of the backbone model can be easily transferred thanks to the positive data augmentation of HOOD, which further broadens the practical usage of HOOD in modern industry.

