# OpenReview forum: "Harnessing Out-Of-Distribution Examples via Augmenting Content and Style"
_ICLR.cc/2023/Conference — ICLR 2023 poster_

### Official Review · Reviewer_sqif · 2022-10-20

**Confidence:** 3
**Correctness:** 3
**Technical Novelty And Significance:** 4
**Empirical Novelty And Significance:** 4
**Recommendation:** 5

**Clarity, Quality, Novelty And Reproducibility:**

The idea of causal disentanglement is novel.
But authors do not conduct experiment on OOD setting or compare with other generalization algorithms.


**Strength And Weaknesses:**

Strength:
1. The proposed method can improve performance on all 3 OOD applications.
2. The causal disentanglement method proposed in this paper can separate content and style, which is novel.

Weakness:
1. According to Algorithm 1, each iteration needs to pre-train the VAE, which consumes a lot of computational cost and running time cost. The author needs to give the time complexity of running the algorithm, and the running time of the experiment, and compare it with the baseline methods.
2. The author only carried out three kinds of OOD applications through experiments, but what is the proposed methods in the traditional OOD setting with supervised learning, i.e., the training set and the test set do not belong to the same distribution. And how does this data augmentation method perform compared to other traditional generalization algorithms, such as ERM, DRO [1] or IRM [2].
3. Why use adversarial data augmentation, the author did not give sufficient reasons and proofs. For example, after we can decouple the content and style, why not use random data augmentation, or other data augmentation methods.
4. Whether the adversarial data augmentation can hurt the performance. For example, adversarial training will degrade the performance of the model on clean image.

[1] Distributionally Robust Neural Networks for Group Shifts: On the Importance of Regularization for Worst-Case Generalization

[2] Invariant Risk Minimization


**Summary Of The Paper:**

To address three applications of OOD: OOD detection, Open-set SSL and Open-set DA, this work proposes a novel data augmentation method, named HOOD. They analyze the data generation process from a causal perspective, and uses a variational inference network to disentangle the content and style from the input data. Then, they perform data augmentation on the content and style, respectively, to obtain benign and malign OOD samples. Finally, the authors use these augmented samples to improve the generalization of the model. Extensive experiments on three OOD applications demonstrate the effectiveness of this method.

**Summary Of The Review:**

Please refer to the “Weakness”.

---

> ### Author Response · Authors · 2022-11-15
> **Response to Reviewer sqif (1/2)**
>
> R4: We thank Reviewer sqif for your helpful feedback. We have carefully compared the computational cost between HOOD and some baseline methods, conducted experiments in the supervised setting, justified our choice of adversarial data augmentation and explained the effect of adversarial training. All the changes are highlighted in green in the revised paper.
>
> *Q1*: Computational cost of HOOD compared to other methods.
> *A1*: Thanks. We want to stress that designing an efficient method with small time complexity is not the intended contribution of HOOD. Therefore, we did not make any claim about efficiency in our paper.
> To give a quantitative comparison, here we provide the running time on 3090 GPU compared to some typical baseline methods:
> * OOD detection:
> | method     | time  |
> |------------|-------|
> | Likelihood | 6.2h  |
> | OpenGAN    | 7.8h  |
> | HOOD       | 11.4h |
> * Open-set SSL:
> | method    | time |
> |-----------|-------|
> | DS3L      | 15.4h |
> | OpenMatch | 10.5h |
> | HOOD      | 13.7h |
> * Open-set DA
> | method | time  |
> |--------|-------|
> | UAN    | 8.5h  |
> | STA    | 9.1h  |
> | HOOD   | 12.4h |
> * It is unsurprising that the proposed HOOD costs more time for training compared to most baseline methods, due to the fact that HOOD has to train two branches for content and style disentanglement. Yet, according to the no-free lunch theorem, given the effectiveness of HOOD on three OOD tasks, the sacrifice on time complexity is reasonable.
>
>
> *Q2*: Result on the supervised setting under distribution shift.
> *A2*: By following the requested setting, we conduct experiments on supervised learning with distribution shift. Specifically, we follow Sagawa et al., 2020 by using CelebA dataset to construct our training dataset and test dataset and compare HOOD to GroupDRO ([D1]) and IRM ([D2]). In this setting, as the distribution index (or group index) is available, we consider different groups as different styles. Moreover, we conduct positive data augmentation to focus on augmenting the data from minor groups. The result is shown below:
> | method  | average test accuracy | worst-group accuracy |
> |----------|---------------|---------------|
> | ERM     | 94.8    | 41.1 |
> | IRM     | 90.5          | 63.4  |
> | GroupDRO | 93.5          | 86.7 |
> | HOOD    | 96.2          | 92.1 |
> * As we can see, HOOD still shows strong performance both on average accuracy and worst-group accuracy.
>   * Intuitively, thanks to the disentanglement, HOOD makes predictions based on the content feature without being influenced by style changing, thus can achieve better averaged test accuracy.
>   * More importantly, the key challenge of distributionally robust supervised learning is that some distributions (or groups) have extremely limited numbers. However, the positive data augmentation of HOOD can largely address such a problem by enriching the minor groups. As a result, HOOD can still perform effectively on the worst-case distributions.
>
> [D1] Sagawa et al., Distributionally Robust Neural Networks for Group Shifts: On the Importance of Regularization for Worst-Case Generalization, ICLR, 2020.
> [D2] Arjovsky et al., Invariant Risk Minimization, arXiv, 2020.

---

> > ### Author Response · Authors · 2022-11-15
> > **Response to Reviewer sqif (2/2)**
> >
> > *Q3*: Justification of using adversarial data augmentation.
> > *A3*: Thanks for pointing that out.
> > * First of all, adversarial data augmentation can directly conduct an intervention on the latent variables without influencing each other, which is perfect for our intuition of augmenting content and style. However, random data augmentation cannot solely change the content or style, and it is not effective for enhancing model robustness.
> > * Secondly, adversarial learning can largely improve the model robustness when facing OOD examples, which has been testified by [D3, D4]. Therefore, we choose adversarial augmentation, which has been highlighted in the revised paper.
> > * Further, we have conducted experiments to compare our adversarial data augmentation to Mixup [D5] and GAN [D6], the results are shown below:
> > | Tasks          | Adversarial (ours)   | Mixup         | GAN              |
> > |--------------- |-------------------- |------------   |------------   |
> > | OOD detection | $86.12\pm0.6$ | $78.45\pm1.2$   | $75.58\pm1.1$    |
> > | Open-set SSL | $83.55\pm1.2$ | $79.65\pm1.3$     | $73.34\pm1.4$    |
> > | Open-set DA | $72.4\pm1.6$ | $68.26\pm1.5$  | $67.74\pm0.8$    |
> >   * For Mixup, we use the example pairs that have identical class predictions to produce benign data and use the pairs that have different class predictions to produce malign data.
> >   * As for GAN, we use the generator to produce real data and fake data which are considered benign data and malign data, respectively.
> > Based on the comparison with other perturbation methods, we find that the adversarial perturbation suits our HOOD the best. Therefore, we can say that using adversarial data augmentation is well-supported and effective.
> >
> > [D3] Wang et al., Improving Out-of-Distribution Generalization by Adversarial Training with Structured Priors, NeurIPS, 2022.
> > [D4] Alhamoud et al., Generalizability of Adversarial Robustness Under Distribution Shifts, arXiv, 2022.
> > [D5] Zhang et al., mixup: Beyond Empirical Risk Minimization, ICLR, 2018.
> > [D6] Goodfellow et al., Generative Adversarial Nets, NeurIPS, 2014.
> >
> >
> > *Q4* Whether adversarial training will hurt the performance.
> > *A4* Thanks, this is also an interesting point.
> > * Firstly, as [D7] reveal: “*when training with few samples, adversarial training has a positive effect on model generalization. However, as training data increase, the standard accuracy of robust models drops below that of the standard model*”, our problem setting only has scarce labeled training data and abundant unlabeled training data. Therefore, using adversarial data augmentation in our paper to improve the test accuracy is consistent with the findings of [D7].
> > * Moreover, adversarial training has been widely used in the fields of semi-supervised learning (VAT, [D8]), domain adaptation (DANN, [D9]), and OOD detection (ODIN, [D10]) to improve generalization ability. Therefore, harnessing OOD data via adversarial data augmentation is well-supported, which is also validated by our ablation study.
> >
> >
> > [D7] Tsipras et al., Robustness May Be at Odds with Accuracy, ICLR, 2019.
> > [D8] Miyato et al., Virtual Adversarial Training: A Regularization Method for Supervised and Semi-Supervised Learning, TPAMI, 2018.
> > [D9] Gannin et al., Unsupervised domain adaptation by backpropagation, ICML, 2015.
> > [D10] Liang et al., Enhancing the reliability of out-of-distribution image detection in neural networks, ICLR, 2018.

---

> ### Author Response · Authors · 2022-11-17
> **Comment for Reviewer sqif**
>
> Dear Reviewer sqif:
>
> We really appreciate your efforts to help improve this paper. We have carefully addressed the mentioned concerns. Are there any remaining problems? We will try our best to address them.
>
> Best,
> Authors.

---

> ### Author Response · Authors · 2022-11-19
> **Further Discussion**
>
> Dear Reviewer sqif:
>
> We want to express our appreciation for your valuable suggestions, which greatly helped us improve the quality of this paper. We are also glad that you agreed that both the technical novelty and the empirical novelty of this paper are significant. Therefore, we have taken our maximum effort into addressing all the mentioned weaknesses and improving the correctness. Your further opinions are very important for evaluating our revised paper and we are hoping to hear from you. Thank you so much!
>
> Best,
> Authors.

---

> ### Comment · Reviewer_sqif · 2022-11-22
> **Keep the score.**
>
> Thanks for your responses. I also agree with Reviewer MQ8c that the causal diagrams is still unconvincing. So I don't change my score.

---

> > ### Author Response · Authors · 2022-12-02
> > **Further response for your concern**
> >
> > Dear Reviewer sqif:
> >
> > Thank you so much for your response. We have provided explanations about the causal diagram and justified our disentanglement in our discussion with Reviewer MQ8c. We wonder whether our response is satisfactory for you. If there are any remaining problems, please let us know, we would be happy for further discussion. Thank you so much for your time!
> >
> > Best,
> > Authors.

---

### Official Review · Reviewer_Jm4f · 2022-10-23

**Confidence:** 3
**Correctness:** 3
**Technical Novelty And Significance:** 3
**Empirical Novelty And Significance:** 3
**Recommendation:** 5

**Clarity, Quality, Novelty And Reproducibility:**

The paper is well organized overall. However, while it uses a lot of notations (e.g., for parameters, or functions), they are not clearly defined before being used and some details are not provided sufficiently for reproducibility. Algorithm 1 could also be improved by directly referring to each input and output component in the main lines.

**Strength And Weaknesses:**

[+] The idea of defining and discriminating benign and malign OOD data based on the disentanglement of content and style is novel and interesting.

[+] The paper presents a clear explanation of how to harness the ODD data with the proposed framework in several OOD problems. Its effectiveness is validated through various experiments.

[-] The authors need to provide a more specific model architecture for HOOD for reproducibility.

[-] It is not clearly presented how well the proposed method disentangles content and style, although its effectiveness is shown in terms of performance at OOD applications. It will help if the authors can provide at least brief validation results on this. Figure 4 is not very informative and quite hard to recognize.

[-] According to Figure 3, the results seem sensitive to the number of augmented samples (benign OOD data). It is not in line with the assumption that benign OOD data can improve model performance.

[Q1] In Algorithm 1, the usage of unlabeled input data is not well represented. Is it a required input or optional, or depends on the target task? Which part of the pseudo-code does it affect?

[Q2] The authors explain that the pseudo-labels can be utilized in the case of unlabeled data, but I am curious about how sensitive the result is depending on how the pseudo-labels are obtained.

[Q3] It will be helpful if the authors can show examples of benign and malign OOD samples using images.

[Q4] The authors chose to use adversarial data augmentation that adds a learnable perturbation. I wonder if there is any other option for this, and whether it’s optimal.

[Q5] I wonder if there are more recent models for comparison (e.g., ODIN was introduced in 2017, and other methods shown in Table 1 are quite traditional, except OpenGAN) and how this one performs in comparison with those.

[Minor]

- The term ID is used without being introduced.
- p.2:  by the blue and red lines in Fig. 1 -> by the green line and ...
- p.9 (right above Figure 4) can effective eliminate


**Summary Of The Paper:**

This paper proposes a new framework that identifies and utilizes benign and malign OOD based on the disentanglement of content and style features. The benign OOD data may contain new styles but hold contents, and so they can help improve style-invariant model training. The malign OODs are vice versa and can confuse OOD applications. The proposed method is designed as a variational inference framework based on a structural causal model. Data augmentation based on this decomposition improves performance in three representative OOD applications.

**Summary Of The Review:**

This paper introduces a novel view and framework for handling OOD instances, which seems interesting and reasonable. However, more comparisons with recent models and a clearer description of the model details would improve the paper further.

---

> ### Author Response · Authors · 2022-11-15
> **Response to Reviewer Jm4f (1/2)**
>
> **R3**: We thank Reviewer Jm4f for your positive opinions. Here we carefully specified detailed settings, empirically validated our disentanglement, explained the sensitivity to augmentation numbers, validated the sensitivity to pseudo label threshold, visualized the augmented data, justified the choice of adversarial data augmentation, and compared with recent OOD techniques. All the changes are highlighted in green in the revised paper.
>
>
> *Q1*: Clarification of the model architecture for reproducibility.
> *A1*: Thanks. Our model architecture is already specified in the implementation details, we have highlighted it in the revised paper. For more details, please see the submitted code.
>
> *Q2*: Validation of the disentanglements between content and style.
> *A2*: Thanks.
> * To empirically show that the content and style are disentangled from each other, we select the latent variables from different content and style categories and use the learned class and domain classifiers for prediction.
>   * Specifically, there are four kinds of input-prediction types: (1) content-class, (2) content-domain, (3) style-class, and (4) style-domain.
>   * Please check the result in Figure 5 from the revised paper. We can see that only content-class and style-domain can make correct predictions, but the rest content-domain and style-class barely show any meaningful relationships. Thus, we can conclude that our disentanglement is effective.
>
>
> *Q3*: The performance is sensitive to augmentation numbers, and why more benign data lead to a performance drop.
> *A3*: Thanks for pointing out, the performance sensitivity caused by different augmentation numbers is an important finding of our experiment.
> * On one hand, a small augmentation number indicates insufficient style change, thus leading to limited style information. Therefore, the performance would unsurprisingly decrease.
> * On the other hand, too much augmentation would significantly increase the difficulty of style classification, further making the disentanglement of content and style extremely hard. Therefore, the degradation of prediction performance is due to the limitation of model capacity, instead of the effect of benign data. Hence, our experimental result is not a violation of the assumption that benign data is helpful. We have explained this discovery in Section 4.5, and a clearer demonstration has been added in the revision paper.
>
> *Q4*: How the unlabeled data is used.
> *A4*: The unlabeled data can be leveraged using pseudo labels in Step 4 from Algorithm 1. We have added the pseudo label noted by $y_i^{ps}$ for unlabeled set $D^u=\{(x_i, y_i^{ps})\}_{i=1}^u$.
>
> *Q5*: How sensitive is the performance to different pseudo-label choices.
> *A5*: Thanks. We follow [C1] to set the pseudo label threshold as 0.95. To investigate the performance of HOOD under different pseudo-label threshold settings, here we show the sensitivity analysis:
> | Threshold | OOD detection | Open-set SSL | Open-set DA |
> |--------------|---------------|--------------|-------------|
> | 0.8          | 85.5          | 83.8         | 71.5        |
> | 0.9          | 85.6          | 84.6         | 72.1        |
> | 0.95         | 86.1          | 86.2         | 72.4        |
> | 0.99         | 85.3          | 82.5         | 71.5        |
> * We can see that a small threshold and a large threshold would both damage the learning performance. Intuitively, a small threshold would include many noisy labels which are misleading for training.
> * On the other hand, a large threshold would limit the exploit on unlabeled data which is vital for enhancing generalization, similar results can be found in many SSL literatures [C2, C3]. Hence, setting the threshold to $0.95$ is appropriate for our model.
>
> [C1] Lee et al., Pseudo-label: The simple and efficient semi-supervised learning method for deep neural networks, ICMLW, 2013.
> [C2] Rizve et al., In defense of pseudo-labeling: An uncertainty-aware pseudo-label selection framework for semi-supervised learning.
> [C3] Oliver et al., Realistic Evaluation of Deep Semi-Supervised Learning Algorithms

---

> > ### Author Response · Authors · 2022-11-15
> > **Response to Reviewer Jm4f (2/2)**
> >
> > *Q6*: Image illustration of benign and malign OOD data.
> > *A6*: We have fixed the previous visualization which is due to setting a too-small perturbation magnitude $\epsilon=2/255$, thus producing unrecognizable visualizations. However, by following [C4], we set the adversarial perturbation magnitude $\epsilon=4.7$, and the results become much clearer. Please refer to Figure 4 in the revised paper. Note that large perturbation cannot be applied in training, as it would seriously damage the accuracy [C4].
> > * In Figure. 4, we can see that different augmentations can be clearly distinguished:
> >   * The content-augmented images have significantly changed objects which do not resemble their original ones at all. As a result, the model constantly makes wrong predictions. Hence, recognizing such images as malign data can help the OOD detection performance;
> >   * The positive-augmented images remain the objects intact but present a style change (such as “cat” becomes blue, and “car” becomes pink). Such images can be identified as benign data which can help train a style-invariant model.
> >
> > [C4] Tsipras et al., Robustness May Be at Odds with Accuracy, ICLR, 2019.
> >
> >
> > *Q7*: Justification for using adversarial data augmentation.
> > *Q7*:
> > * First of all, adversarial data augmentation can directly conduct an intervention on the latent variables without influencing each other, which is perfect for our intuition of augmenting content and style.
> > * Secondly, adversarial learning can largely improve the model robustness when facing OOD examples, which has been testified by [C5, C6].
> > * Further, we have conducted experiments to compare our adversarial data augmentation with Mixup [C7] and GAN [C8], the results are shown below:
> > | Tasks          | Adversarial (ours)   | Mixup         | GAN              |
> > |--------------- |-------------------- |------------   |------------   |
> > | OOD detection       | $86.12\pm0.6$          | $78.45\pm1.2$    | $75.58\pm1.1$      |
> > | Open-set SSL   | $83.55\pm1.2$          | $79.65\pm1.3$    | $73.34\pm1.4$      |
> > | Open-set DA        | $72.4\pm1.6$           | $68.26\pm1.5$    |$ 67.74\pm0.8 $     |
> >   * For Mixup, we use the example pairs that have identical class predictions to produce benign data and use the pairs that have different class predictions to produce malign data.
> >   * As for GAN, we use the generator to produce real data and fake data which are considered benign data and malign data, respectively.
> > Based on the comparison with other perturbation methods, we find that the adversarial perturbation suits our HOOD the best. Therefore, we can say that using adversarial data augmentation is well-supported and effective.
> >
> > [C5] Wang et al., Improving Out-of-Distribution Generalization by Adversarial Training with Structured Priors, NeurIPS, 2022.
> > [C6] Alhamoud et al., Generalizability of Adversarial Robustness Under Distribution Shifts, arXiv, 2022.
> > [C7] Zhang et al., mixup: Beyond Empirical Risk Minimization, ICLR, 2018.
> > [C8] Goodfellow et al., Generative Adversarial Nets, NeurIPS, 2014.
> >
> >
> > *Q8*: Comparison with recent OOD detection methods.
> > *A8*: Thanks for this advice, we have compared our method with POEM [C9], ReAct [C10], and Energy [C11], and the results are shown below:
> > | OOD datasets | LSUN | DTD | CUB | Flowers | Caltech | Dogs |
> > |--------------|------|-----|-----|---------|---------|------|
> > | Energy       | $73.72\pm0.7$ | $72.56\pm0.3$ | $73.16\pm0.5$ | $70.78\pm0.3$ | $79.64\pm0.5$ | $81.56\pm0.6$ |
> > | ReAct        | $82.36\pm0.5$ | $81.45\pm1.0$ | $82.20\pm0.5$ | $79.82\pm0.8$ | $85.66\pm 0.7$ | $88.78\pm0.6$ |
> > | POEM              | $78.56\pm0.6$ | $76.87\pm0.6$ | $79.46\pm0.5$ | $77.50\pm0.6$ | $82.66\pm0.4$ | $83.78\pm0.6$ |
> > | HOOD         | $86.12\pm0.6$ | $83.64\pm0.5$ | $83.53\pm0.6$ | $81.56\pm0.8$ | $87.24\pm0.8$ | $90.86\pm0.6$ |
> > * As shown above, the proposed HOOD can still outperform some of the most recent progress in the OOD detection setting.
> > * Compared to these methods which normally conduct detection based on all the extracted features, our HOOD focuses on capturing the content and leaving out the style noise, thus can achieving improved detection results.
> >
> > [C9] Ming et al., POEM: Out-of-Distribution Detection with Posterior Sampling, ICML, 2022.
> > [C10] Sun et al., ReAct: Out-of-distribution Detection with Rectified Activations, NeurIPS, 2021.
> > [C11] Liu et al., Energy-based out-of-distribution detection, NeurIPS, 2020.
> >
> > *Minor*: Terms and typos.
> > *A*: Thanks for pointing this out, we have fixed all errors in the revised paper, which are highlighted in green.

---

> ### Author Response · Authors · 2022-12-02
> **Further Discussion**
>
> Dear Reviewer Jm4f:
>
> We appreciate your elaborative work and constructive comments. We have tried our best to address each of your concerns. However, we found that your opinion of our paper got more negative, which is quite confusing to us. Could you please tell us what part of our response is unclear to you so that we can provide a further explanation? Thank you so much!
>
> Best,
> Authors.

---

### Official Review · Reviewer_MQ8c · 2022-10-24

**Confidence:** 4
**Correctness:** 3
**Technical Novelty And Significance:** 2
**Empirical Novelty And Significance:** 2
**Recommendation:** 5

**Clarity, Quality, Novelty And Reproducibility:**

+ The paper is easy to follow
+ The idea of disentangling the content and the style is not new, and has been widely discussed, see the above references.
+ Also, using variational inference to estimate them is a common way in the community.
+ The submission seemingly does not contain enough details to reproduce the results.


**Strength And Weaknesses:**

**Strength:**

The problem studied in this paper is quite important to the community. The motivation is convincing and the logic is easy to follow. The paper is generally well written.

 **Weaknesses:**

+ My main concern is about the causal diagrams presented in Fig. 1.
   - In Fig. 1a, the authors assume that the style $S$ causes the environment/domain $D$. Apparently, it contradicts the common assumption widely used in the literature that the environment $D$ causes the style $S$, *e.g., Magliacane et al., 2018; Suter et al., 2019; Wang and Jordan, 2021; Sun et al., 2021; Lu et al., 2022; Quinzan et al., 2022*. Why do the authors make this assumption? Is there any particular reason for this? How practical is it in comparison with the common assumption?
   - Also, in Fig. 1a, the authors do not assume any dependence between $S$ and $C$, which is quite limited in real world scenarios. One commonly spurious correlation between $S$ and $C$ is due to the existence of the environment $D$, see *Sun et al., 2021; Liu et al., 2021; von Kugelgen et al., 2021*. In other words, we usually have the spurious path $S \leftarrow D \rightarrow C$, which also contradicts the assumed causal diagram in Fig. 1b.
   -  In fact, the caption of Fig. 1a is incorrect. Note that, a SCM is defined as a set of functional assignments accompanying with its corresponding causal diagram. Hence, Fig. 1a is only a causal diagram, no a SCM.
   - In Fig. 1c, both causal edges and non-causal edges are mixed up, which could be misleading to readers.

+ Another main concern is about the identifiability. That is, how is it guaranteed, both theoretically and practically, that the content can be distinguished from the style only from the observed data? This is one of the key problems in the thread of work on causal disentanglement, *e.g., Arjovsky et al., 2019; Sun et al., 2021; von Kugelgen et al., 2021*.

+ From the experimental results, it seems that the performance improvement is small, most only having around 1% improvement.

+ notation abusing, e.g., the capitals are used in Fig. 1, whilst the letters are used in the text, etc.
+ unexplained terms, e.g.,
  - what do you mean by ID data? There exists no explanation before it first appears in the paper.
  - no explanation on the open-set methods in the introduction.
+ grammar errors, e.g.,
  - the first sentence in the abstract contains two independent clauses, but without a conjunction, etc.
  - on line 6 of the abstract, the indefinite article "a" is missed before "HOOD method".
  - on line 4, "deep models meet with domain shift ..." should be replaced with "deep models meet domain shift"?

**References**

- Suter et al., *Robustly disentangled causal mechanisms: Validating deep representations for interventional robustness.* ICML 2019.
- Wang and Jordan. *Desiderata for representation learning: A causal perspective.* 2021.
- Quinzan et al., *Learning counterfactually invariant predictors.* 2022.
- Magliacane et al., *Domain adaptation by using causal inference to predict invariant conditional distributions.* NeurIPS 2018.
- Lu et al., *Invariant causal representation learning for out-of-distribution generalization.* ICLR 2022.
- Sun et al., *Recovering Latent Causal Factor for Generalization to Distributional Shifts.* NeurIPS 2021.
- Liu et al., *Learning causal semantic representation for out-of-distribution prediction.* NeurIPS 2021.
- Arjovsky et al., *Invariant risk minimization.* 2019.
- von Kugelgen et al., *Self-supervised learning with data augmentations provably isolates content from style.* NeurIPS 2021.

**Summary Of The Paper:**

The paper proposes a new approach called HOOD to disentangling the content and style features, and then basing on them, design a data augmentation method to construct benign and malign OOD data. They validate the proposed approach on the tasks of OOD detection, open-set SSL, and open-set DA.

**Summary Of The Review:**

My main concerns are on the assumptions over the causal diagrams and on the identifiability, and the experimental results also seems not quite impressive.

---

> ### Author Response · Authors · 2022-11-15
> **Response to Reviewer MQ8c (1/2)**
>
> **R2**: We thank Reviewer MQ8c for your constructive comments. We have carefully justified our assumptions, fixed all the terminologies and expressions, and provided empirical evidence for the disentanglement guarantee. All the changes are highlighted in green in the revised paper.
>
> We want to stress that:
> 1) Although the disentanglement of content and style is common in the field of causality, such a technique has never been leveraged to explain and tackle various OOD problems. However, our novelty lies in the flexibility of HOOD in tackling different types of OOD applications.
> 2) Additionally, existing disentanglement works only focus on the separation, but our work has provided a feasible way to further improve learning by intervening the separated variables.
> Such two aspects can be a valid contribution to both the OOD generalization community and the causality community.
>
>
> *Q1*: Justification of the assumption that style $S$ (content $C$) causes domain $D$ (class $Y$). The practical meaning of such an assumption.
> *A1*: Thanks for pointing this out.
> * First, we want to clarify that our assumption is not a contradiction to common sense. Intuitively, the domain is actually a human-defined concept that is based on the identified style. For example, most well-known domain adaptation datasets are directly downloaded from the internet where different style types are mixed together. The dataset builder has to annotate them with a domain label based on their style features.
> * Moreover, collecting data from various environments to obtain different styles is expensive and unrealistic. Therefore, in our experiments, we follow [B2, B4] by using data augmentation to obtain different styles. During this process, each data augmentation only introduces a specific style, then we manually assign a domain label based on the style type. Hence, in this scenario, assuming that the style causes the domain label is more appropriate than the mentioned one.
> * Furthermore, there are two general assumptions, despite the mentioned assumption that environment $D$ (class $Y$) causes the style $S$ (content $C$), there are many works such as [B1, B2, B3] assume the style $S$ (content $C$) is the cause, and the environment $D$ (class $Y$) is the effect. Therefore, assuming latent variables cause the observations is well-supported.
>
> [B1] Liu et al., Learning Causal Semantic Representation for Out-of-Distribution Prediction, NeurIPS, 2021.
> [B2] Mitrovic et al., Representation Learning via Invariant Causal Mechanisms, ICLR, 2021.
> [B3] Zhang et al., CausalAdv: Adversarial Robustness through the Lens of Causality, ICLR, 2022.
> [B4] von Kugelgen et al., Self-Supervised Learning with Data Augmentations Provably Isolates Content from Style, NeurIPS, 2021.
>
>
> *Q2*: Justification of the independence assumption between $S$ and $C$.
> *A2*:
> * Firstly, Fig. 1 (a) is not a diagram for real-world scenarios. Instead, it is the **ideal** data generation process, in which the content and style should not be influenced by each other. We have made this clear in the revision paper.
> * Secondly, we already considered the spurious correlation between $S$ and $C$ as the erroneous estimation, which is shown by the dashed lines in Fig. 1 (b).
> * More importantly, one of our main contributions is to design a disentanglement method to break these spurious paths, such that an invariant model can be learned.
>
> *Q3*: Incorrect definition of Fig. 1 (a).
> *A3*: Thanks, we have changed our definition of Fig. 1 (a) to “causal diagram” in the revised version.
>
> *Q4*: Mixed causal and non-causal edges in Fig. 1 (c).
> *A4*: We have clarified the difference between causal edges and non-causal edges (intervention process) by using solid lines and dashed lines, respectively.

---

> > ### Author Response · Authors · 2022-11-15
> > **Response to Reviewer MQ8c (2/2)**
> >
> > *Q5*: Identifiability of the proposed method, and how the disentanglement of content and style are guaranteed.
> > *A5*:
> > * We would like to emphasize that identifiability is not the focus of this paper which commonly requires strong assumptions and is quite challenging. Although requesting identifiability is not realistic for most of the existing variational frameworks, our framework leverages the general VAE framework which can be easily extended to the most recent progress [B5, B6].
> > * To empirically show that the content and style are disentangled from each other, we select the latent variables from different content and style categories and use the learned class and domain classifiers for prediction.
> >   * Specifically, there are four kinds of input-prediction types: (1) content-class, (2) content-domain, (3) style-class, and (4) style-domain.
> >   * Please check the result in Figure 5 from the revised paper. We can see that only content-class and style-domain can make correct predictions, but the rest content-domain and style-class barely show any meaningful relationships. Thus, we can conclude that our disentanglement is effective.
> >
> > [B5] Kong et al., Partial Identifiability for Domain Adaptation, ICML, 2022.
> > [B6] Gulrajani et al., Identifiability Conditions for Domain Adaptation, ICML, 2022.
> >
> > *Q6*: Limited experimental improvements.
> > *A6*:
> > * We want to point out that the three OOD applications are highly competitive fields, but our method is shown to be more effective than sophisticated competitors on most datasets. For example, in Open-set SSL on CIFAR10 dataset, our improvement on corrupted acc is more than $3$%, and in Open-set DA on VisDA data, our improvement is more than $6$%, both of which are nontrivial improvements.
> > * More importantly, there is no existing method that has the ability to tackle all three OOD applications, but the proposed HOOD shows both flexibility and effectiveness on these important problems, which can be a valid contribution to the community.
> >
> > *Q7*: Notation difference between figures and text; Unexplained terms; and Grammar errors.
> > *A7*: Thanks for the notification. We have fixed the notations to make sure consistency between the figures and text. The terms and typos have been fixed in the revised paper. As for the request to introduce open-set problems, we think this field is more appropriate to be considered as related works, which has been highlighted in the related work section.

---

> ### Author Response · Authors · 2022-11-17
> **Comment for Reviewer MQ8c**
>
> Dear Reviewer MQ8c:
>
> Thanks a lot for your efforts in reviewing this paper. We tried our best to address the mentioned concerns. Are there unclear explanations or remaining problems? We will try our best to address them.
>
> Best,
> Authors.

---

> ### Author Response · Authors · 2022-11-19
> **Further Discussion**
>
> Dear Reviewer MQ8c:
>
> We really appreciate you for taking the time to review this paper. As we have carefully checked your comments again, we have made sure that all your concerns are responded to thoroughly. Thanks to these constructive suggestions, our paper has been largely improved. We are looking forward to hearing your further opinion on our vision paper. Thank you very much!
>
> Best,
> Authors.

---

> ### Author Response · Authors · 2022-12-02
> **Any more concerns?**
>
> Dear Reviewer MQ8c:
>
> We really appreciate you for spending time on our submission. We have tried our best to address each of your concerns, if there is any explanation that does not address your concerns, please let us know. Moreover, if there are any remaining problems, we would be happy for further discussion. Thanks again for providing insightful suggestions which have greatly helped improve our paper, we are hoping to hear your further opinions.
>
> Best,
> Authors.

---

### Official Review · Reviewer_btZ1 · 2022-10-25

**Confidence:** 3
**Clarity, Quality, Novelty And Reproducibility:** his paper is good enough in all four …
**Correctness:** 4
**Technical Novelty And Significance:** 4
**Empirical Novelty And Significance:** 4
**Recommendation:** 6

**Strength And Weaknesses:**

This paper is well-written and interesting. The formulations used in this paper are novel. Various organized experiments show that the proposed algorithm works practically well in various situations and datasets and, in most cases, shows SOTA performance. It was good to show the effect of using benign and malign samples in an ablation study.

Despite the many advantages of this paper mentioned earlier, I am puzzled by the qualitative results of data augmentation. According to Figure 4, despite the data augmentation, the image content does not seem to change significantly between positive and negative. If more qualitative results were provided, it would be better to understand the operational consequences of data augmentation. Also, I wonder if the augmentation algorithm will work well for higher-resolution images (e.g., ImageNet image examples).

Comparing the effects of two data augmentation methods not covered in this paper (maintaining both content and style or moving them away) would also be helpful in an ablation study.

Simple error: In chapter 3.1, on page 3, the operator for a conditional independent is duplicated twice.

**Summary Of The Paper:**

In this paper, the authors propose a data augmentation method that separates the content and style of images to solve the OOD (out-of-distribution) problem. To this end, the content and style features of the image are causally separated. By keeping one of the two features similar to the original image and widening the distance of the other features, benign OOD and malign ODD data are generated, respectively. Experimental results show that the proposed algorithm is applicable to various OOD problems.

**Summary Of The Review:**

Overall, I read this paper as interesting and positive for acceptance. Adding some qualitative and quantitative experiments may further improve the quality of this paper.

---

> ### Author Response · Authors · 2022-11-15
> **Response to Reviewer btZ1**
>
> **R1**: We thank Review btZ1 for your positive opinion. We have carefully provided a clear visualization of the augmented data, complemented the ablation study, and fixed the typos. All the changes are highlighted in green in the revised paper.
>
>
> *Q1*: Insignificant change of positive and negative data augmentations. More qualitative results of data augmentation.
> *A1*:
> * Thanks. The insignificant change is due to choosing a too-small perturbation magnitude, thus producing unrecognizable visualizations. However, by following [A1], we set the adversarial perturbation magnitude $\epsilon=4.7$ (previously set to $2/255$), the augmentation results become much clearer. Please refer to Figure 4 in revised paper. Note that large perturbation cannot be applied in training, as it would seriously damage the accuracy [A1].
> * In Figure. 4, we can see that different augmentations can be clearly distinguished:
>   * The negative-augmented images have significantly changed objects which do not resemble their original ones at all. As a result, the model constantly makes wrong predictions. Hence, recognizing such images as malign data can help the OOD detection performance;
>   * The positive-augmented images remain the objects intact but present a style change (such as “cat” becomes blue, and “car” becomes pink). Such images can be identified as benign data which can help train a style-invariant model.
>
> [A1] Tsipras et al., Robustness May Be at Odds with Accuracy, ICLR, 2019.
>
>
> *Q2*: Effect on higher-resolution images.
> *A2*: Thanks. We have provided examples from ImageNet30 dataset with resolution $256\times256$ in Fig. 6 from the appendix. We can see the similar phenomenon as the CIFAR10 examples, which demonstrates the effectiveness of our data augmentation on high resolution images.
>
> *Q3*: Ablation study on maintaining both content and style or moving them away.
> *A3*: Thanks for pointing this out. Here we denote maintaining both content and style as “w/o both augmentations”. Moreover, moving content and style away is as done by our HOOD which augments both content and style. The ablation study is shown below:
> | Application            | OOD detection | Open-Set SSL | Open-Set DA |
> |------------------------|---------------|--------------|-------------|
> | w/o disentanglement    | $84.94\pm1.3$    | $82.55\pm1.1$   | $64.6\pm0.9$   |
> | w/o both augmentations | $80.83\pm0.8$    | $81.14\pm1.2$   | $64.4\pm1.2$   |
> | w/o benign OOD data    | $85.95\pm1.8$    | $83.32\pm2.0$   | $66.3\pm2.5$   |
> | w/o malign OOD data    | $82.50\pm2.2$    | $85.40\pm0.8$   | $71.8\pm1.2$   |
> | HOOD                   | $86.12\pm0.6$    | $86.22\pm2.7$   | $72.4\pm1.6$   |
> * We can see that augmenting both content and style shows performance degradation compared to both “w/o benign OOD data” and “w/o malign OOD data”, which is not surprising as both of our augmentation modules are effective.
> * Additionally, we found that our data augmentation is more important than disentanglement, as training without both augmentations shows performance drop when compared to training without disentanglement.
> * Therefore, augmenting content and style is essential for handling OOD examples and maintaining both content and style would hinder the ability to handle OOD examples.
> We have complemented this part in Section 4.5 from the revised paper.
>
> *Q4*: Typo of conditional independence.
> *A4*: Thanks for pointing this out, we have fixed it in the revised version.

---

> > ### Comment · Reviewer_btZ1 · 2022-12-08
> > **Keep the first recommendation**
> >
> > Thanks to the authors for their responses.
> > The authors have addressed my concerns in the revised manuscript.
> > Also, after checking the concerns and discussions of the other PCs, this manuscript is still above the accepted borderlines.

---

> > > ### Author Response · Authors · 2022-12-08
> > > **Re recommendation**
> > >
> > > Dear reviewer btZ1:
> > >
> > > Thank you for taking the time to review this paper. Your comments are sincerely appreciated and have helped improve this manuscript.
> > >
> > > Best,
> > > Authors.

---

### Author Response · Authors · 2022-11-15
**General Response**

We thank the reviewers for their insightful and constructive reviews of our manuscript. Delightfully, we are glad that the reviewers found that:
* Our core idea is novel and interesting, the motivation is well-supported (Reviewers btZ1, Jm4f, and sqif),
* Results on various OOD applications are effective (Reviewers btZ1, Jm4f, and sqif),
* Writing is easy to follow (Reviewers btZ1, MQ8c, Jm4f, and sqif).

Based on all the comments from the Reviewers, here we provide a general response to the concerns raised by multiple reviewers. The individual responses are commented on below each review.
* Regarding the visualization of our data augmentation (Reviewers btZ1 and Jm4f):
  * We have provided the results by setting a large perturbation magnitude such that the visualization results can be more distinguishable.
  * The visualization on higher resolution images is also provided in the Appendix.

* Regarding the guarantee of disentanglement (Reviewers MQ8c and Jm4f):
  * We have conducted additional experiments to show that our content feature is only linked to the class prediction, and the style feature is only linked to the domain prediction.

* Regarding the justification of using adversarial data augmentation (Reviewers Jm4f and sqif):
  * We have explained that the adversarial manner can solely change the content and style without influencing each other.
  * Comparisons with other popular augmentation methods are provided to show that adversarial data augmentation is a better choice.

We have revised our draft according to your valuable comments. Major revisions are highlighted in green. We sincerely thank all the reviewers. Please feel free to let us know if further details/explanations are helpful.

Best,
Authors.

---

### Decision · Program_Chairs · 2023-01-20

**Decision:**

Accept: poster

**Justification For Why Not Higher Score:**

I launched the internal discussion among reviewers. Only one reviewer had deep discussion with me, who increased the score from 3 to 5. Other reviewers did not reply to my comments.

**Justification For Why Not Lower Score:**

One reviewer has a major concern on the causal graph, who provides some references to support that the style determines the domain, which conflicts with the assumption in this paper that the domain determines the style. In my opinion, I support both. There is no ground truth in this causal graph. Both sides have some supporting evidence (The authors also provide some in their responses). Moreover, it is beneficial to speak different voices to thrive in the community.

This paper is solid in the motivation, philosophy, technique and experiments. Based on my experience and criteria, this paper deserves the acceptance.

**Metareview: Summary, Strengths And Weaknesses:**

Based on the collected information from all reviewers and my personal judgment, I can make the recommendation on this paper, **accept**. Here are the comments that I summarized, which include my opinion and evidence.

**Research Problem**

This paper studies the well-defined Out-of-Distribution (OOD) problem. The authors argue that the OOD samples can be further divided into two categories. Based on that, the authors aim to explore the mechanisms of benign and malign OOD data. Specifically, the authors claim that the benign and malign OOD data can be identified from the content and style, where the content means the categorical information and the style includes other influential factors such as brightness, orientation, and color (I believe the authors put all non-content information into this category). In light of this, the authors propose an algorithm named HOOD to achieve this. Furthermore, the authors conduct data augmentation to generate the benign and malign OOD data, where the benign OOD data contain novel styles but hold our interested contents, in contrast, the malign OOD data inherit unknown contents but carry familiar styles.

**Presentation**

The presentation is good.

**Motivation**

The motivation of this paper sounds reasonable to me. The OOD data are intuitively harmful to the learning task. The authors provide some literature to demonstrate the existence of benign OOD samples. Putting the above together, it is reasonable to categorize the ODD data into two categories.
Further, the authors conduct causal disentanglement from the content and style aspects. The major concern from several reviewers is the causal graph. One reviewer provides some references to support that the style determines the domain, which conflicts with the assumption in this paper that the domain determines the style. In my opinion, I support both. There is no ground truth in this causal graph. Both sides have some supporting evidence (The authors also provide some in their responses). Moreover, it is beneficial to speak different voices to thrive in the community.

**Philosophy**

The philosophy of this paper can be further strengthened. For example, why disentangle the image from the content and style aspects? It seems that the authors put all non-content information into this category. Do you have the third aspect? Some discussions and explanations are needed.

Moreover, the authors assume that the benign OOD data contain novel styles but hold our interested contents, in contrast, the malign OOD data inherit unknown contents but carry familiar styles. This point also needs discussion and empirical verification (Please refer to my points in the experiment section).


**Technique**

The technique of this paper consists of two parts. One is to disentangle the samples from the content and style aspects. Another is to conduct data augmentation to generate benign and malign OOD data. The authors use both the generated benign and malign samples to boost the performance. Specifically, the benign samples increase the training volume, while the malign samples work as a constraint via a one-vs-all classifier.

**Experiments**

The experimental results are rich and solid in various settings. The authors provide fair comparisons and in-depth analysis of their proposed method as well.

The authors provide the execution time in the response. I encourage the authors to put it in the appendix.

The authors need to verify whether the samples with unknown contents but carrying familiar styles are malign OOD. It is suggested to include these malign samples in training and expect performance degradation. In the current version, the authors only verify the impact of benign samples.

**Minor**

1. In ‘OOD data$^1$.’, the superscript comes after the period.

2. Some figures occupy too many top margins.

**Note From Pc:**

if the above contains the word "oral" or "spotlight" please see: "oral" presentation means -> notable-top-5% and "spotlight" means -> notable-top-25%. As stated in our emails, we are disassociating presentation type from AC recommendations

**Summary Of Ac-Reviewer Meeting:**

This paper has the mixed scores from the reviewers. The reviewers did not actively participate in the discussion or message/email reply, which did not make the zoom meeting happen.